# Biodiversity of ecosystems in an arid setting: The late Albian plant communities and associated biota from eastern Iberia

Eduardo Barrón[1]*, Daniel Peyrot[2], Carlos A. Bueno-Cebollada[1], Jiří Kvaček[3], Sergio Álvarez-Parra[4,5], Yul Altolaguirre[6], Nieves Meléndez[7]

**1** Museo Geominero, Centro Nacional Instituto Geológico y Minero de España CN-IGME CSIC, Madrid, Spain, **2** School of Earth and Environment, Centre for Energy Geoscience, University of Western Australia, Crawley, Western Australia, Australia, **3** National Museum Prague, Praha, Czechia, **4** Departament de Dinàmica de la Terra i de l'Oceà, Facultat de Ciències de la Terra, Universitat de Barcelona, Barcelona, Spain, **5** Institut de Recerca de la Biodiversitat (IRBio), Universitat de Barcelona, Barcelona, Spain, **6** OCEEH Research Center 'The role of cultura in early expansions of humans', Heidelberg Academy of Sciences, Senckenberg Research Institute, Frankfurt am Main, Germany, **7** Deparment of Geodinámica, Estratigrafía y Paleontología, Facultad de Ciencias Geológicas, Universidad Complutense de Madrid, Madrid, Spain

☙ These authors contributed equally to this work.
* e.barron@igme.es

**Data Availability Statement:** All relevant data are within the paper and its Supporting Information files.

## Abstract

Deserts are stressful environments where the living beings must acquire different strategies to survive due to the water stress conditions. From the late Albian to the early Cenomanian, the northern and eastern parts of Iberia were the location of the desert system represented by deposits assigned to the Utrillas Group, which bear abundant amber with numerous bioinclusions, including diverse arthropods and vertebrate remains. In the Maestrazgo Basin (E Spain), the late Albian to early Cenomanian sedimentary succession represents the most distal part of the desert system (fore-erg) that was characterised by an alternation of aeolian and shallow marine sedimentary environments in the proximity of the Western Tethys palaeo-coast, with rare to frequent dinoflagellate cysts. The terrestrial ecosystems from this area were biodiverse, and comprised plant communities whose fossils are associated with sedimentological indicators of aridity. The palynoflora dominated by wind-transported conifer pollen is interpreted to reflect various types of xerophytic woodlands from the hinterlands and the coastal settings. Therefore, fern and angiosperm communities abundantly grew in wet interdunes and coastal wetlands (temporary to semi-permanent freshwater/salt marshes and water bodies). In addition, the occurrence of low-diversity megafloral assemblages reflects the existence of coastal salt-influenced settings. The palaeobotanical study carried out in this paper which is an integrative work on palynology and palaeobotany, does not only allow the reconstruction of the vegetation that developed in the mid-Cretaceous fore-erg from the eastern Iberia, in addition, provides new biostratigraphic and palaeogeographic data considering the context of angiosperm radiation as well as the biota inferred in the amber-bearing outcrops of San Just, Arroyo de la Pascueta and La Hoya (within Cortes de Arenoso succesion). Importantly, the studied assemblages include *Afropollis*, *Dichastopollenites*, *Cretacaeiporites* together with pollen produced by Ephedraceae

**Funding:** This study is a contribution to the project CRE CGL2017-84419 AEI/FEDER, UE from the Ministerio de Ciencia, Innovación y Universidades (Spain) and the "Severo Ochoa" extraordinary grants for excellence IGME-CSIC (AECEX2021). The coauthor S.Á.-P. thanks the support from the Secretaria d'Universitats i Recerca de la Generalitat de Catalonia (Spain) and the European Social Fund (2021FI_B2 0003). The funders had no role in study design, data collection and analysis, decision to publish, or preparation of the manuscript.

**Competing interests:** The authors have declared that no competing interests exist.

(known for its tolerance to arid conditions). The presence of these pollen grains, typical for northern Gondwana, associates the Iberian ecosystems with those characterising the mentioned region.

## Introduction

The Albian vegetation of western Europe has been studied in several works which described micro-, meso- and macroremains [1–7], and highlighting the prevalence of ferns and gymnosperms–mainly conifers (Araucariaceae-Podocarpaceae-Pinaceae-Cupressaceae-Cheirolepidiaceae)–, and the increasing representation of angiosperms. The distinctiveness of the western European flora has been recognised for a long time and several biogeographic provinces have been named on the basis of the nature of the fossil assemblages (i.e., palynological and megaremains [8, 9]). The increased attention to the time-interval is justified by the rising of the dominance of the flowering plants in many ecosystems [10–13] including arid environments [14]. During the later part of the Early Cretaceous, SW Europe has been described as one of the probable migratory routes of angiosperms originating from a northern Gondwanan centre of radiation [15]. The successive northward migrations of angiosperms, well-characterised by both pollen and megaremain records, suggest broadly similar climate conditions in low to mid-latitude settings (i.e., low latitudinal climatic gradient) or selective migration of taxa able to tolerate a wide range of climatic conditions [16]. During most of the Albian, Iberia was part of a climatic zone referred to as the Northern Hot Arid (NHA) belt [17], where a desert system (erg) developed over an extensive area [18].

Arid environments, although less represented in fossil record, existed during the Mesozoic. Semi-arid environments are, e.g., reconstructed for the Early Triassic of northeastern Spain and the Upper Jurassic Morrison Formation based on presence of aeolian sandstones [19, 20]. Arid environments were inferred for Early Cretaceous floras of Quedlinburg (Germany) and Las Hoyas (Central Spain) based on occurrence of dwarf lycopod *Nathorstiana* and fern *Weichselia* [21, 22]. Reconstructions of Cretaceous continental ecosystems of Japan based on stable oxygen and carbon isotopes also suggest semi-arid conditions [23]. In addition, aeolian sandtones provide direct evidence of a desert environment during the mid-Cretaceous in mid-to low latitude Asia [24].

In Iberia, the eastern coastal margin consisted of a complex set of depositional environments comprising multifaceted vegetation types including conifer forests [14, 25]. Palaeobotanical studies of ancient desert successions (see i.e. [26, 27]) report plant remains usually not preserved in sand-sized lithologies, which in turn constitute the bulk of their deposits. Assemblages belonging to this type of setting usually include remains that have experienced transport over varying, and sometimes considerable, distances to the site of burial. Considering this limitation and exploiting the characteristics of sets of fossils with distinct taphonomic histories (i.e., megaremains with limited transport vs. miospores disseminated over wider distances), the present work focuses on the reconstruction of the mid-Cretaceous plant communities that developed in the distal part of the desert system (or fore-erg) in the Maestrazgo Basin (Eastern Iberian Ranges, E Spain), considering specially the conifer-dominated ones. While the previous studies carried out in Albian deposits from the Maestrazgo Basin considered palaeobotanical and palynological data separately [28–37], this study represents the first attempt to integrate both data sets into a unified framework, aiming at the characterisation of the arid palaeoenvironmental and their link with the occurrence of amber.

The Cretaceous amber-bearing outcrops from the Maestrazgo Basin have been known for a long time [38] and were traditionally associated with the Escucha Formation [39], an Aptian–

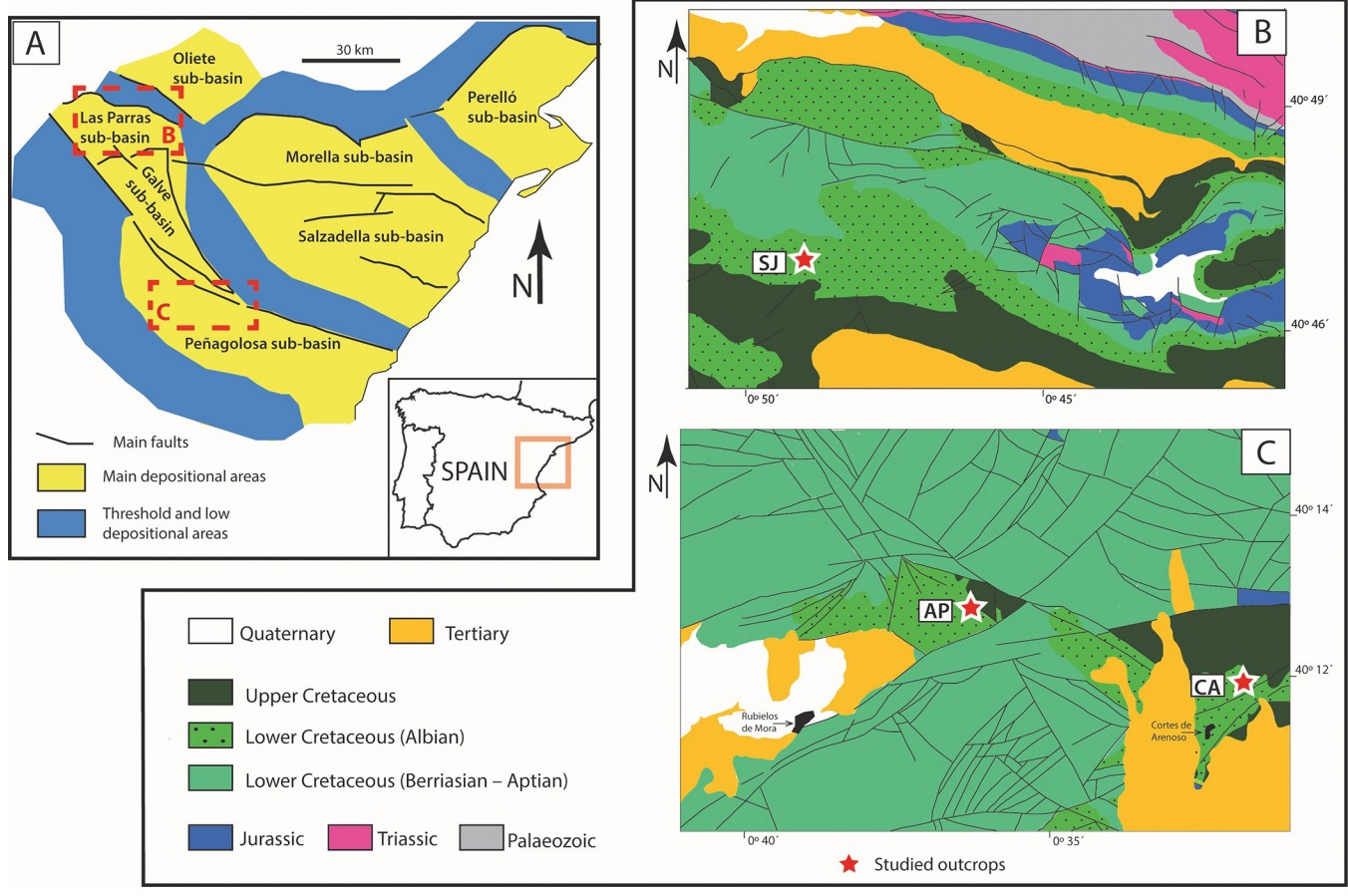

**Fig 1.** Location of the Maestrazgo Basin and its seven sub-basins (A) based on Aurell et al. [45]. The studied locations are indicated by coloured stars (San Just, SJ; Arroyo de la Pascueta, AP; Cortes de Arenoso, CA). (B) Detail of the geological map with the location of San Just (SJ) section in the Las Parras Sub-basin. Based on data from the geological map of Spain Magna 1:50000 [46]. (C) Detail of the geological map showing the location of Arroyo de la Pascueta (AP) and Cortes de Arenoso (CA) sections. Based on data from the geological map of Spain Magna 1:50000 [47].

early Albian unit consisting of coal-bearing deposits [40] which sometimes also present amber [41]. Although more than 30 amber outcrops have been reported in the basin, the sites related to the desert system revealing the most diverse sets of bioinclusions and plant remains [29, 37, 42–44] are the successions of Arroyo de la Pascueta, San Just (both located in Teruel Province) and Cortes de Arenoso (Castellón Province) (Fig 1). The present study is the first to provide a comprehensive treatment of the palaeobotanical and palynological content of the three afore-mentioned mid-Cretaceous sites. This paper aims to: (i) document the micro- and macrofloral assemblages together with the entomological content recovered from highly significant amber-bearing outcrops; (ii) reassess the age of the successions based on the presence of key biostrati-graphic markers; (iii) reconstruct the corresponding ecosystems with a special focus on the plant communities and insects which thrived on them; and (iv) compare the vegetation with coeval floras from presumably wetter settings.

## Material and methods

### Geological setting

The study area is located in the Maestrazgo Basin, on the eastern part of a palaeogeographic domain referred to as the Iberian Basin Rift System (IBRS), Spain (Fig 1A). The sedimentary

succession was deposited during the transition from the Late Jurassic–Early Cretaceous syn-rift stage to the Late Cretaceous post-rift stage, associated with the gradual opening of the Western Tethys and the North Atlantic Ocean [45, 48, 49]. The Maestrazgo Basin forms a NW-SE-oriented depositional trough (Fig 1A), where the sedimentary succession generally becomes thicker towards the SE (Tethyan margin). During the Early Cretaceous the syn-rift sedimentary succession was dominated by thick, continental to coastal siliciclastic-dominated deposits (previously referred to as 'Weald facies') and marine carbonates ('Urgonian facies') [50–55] (Fig 2). Towards the end of the syn-rift stage (uppermost Aptian to early Albian), the sedimentary succession became more homogeneous across the basin, and coal-bearing strata attributed to the Escucha Formation were deposited [56–59]. The Escucha Formation has been interpreted as lagoons and marshes [59, 60].

The Escucha Formation is overlain by strata forming the Utrillas Group, a highly diachronous unit widespread along the IBRS, which has been traditionally interpreted as the transition into the post-rift stage [57, 59, 61]. In the Maestrazgo Basin, the Utrillas Group spanned the middle Albian–earliest Cenomanian [59]. In the Maestrazgo Basin, this Group includes facies originally interpreted as tidally-influenced fluvial deposits [57, 62] and later reinterpreted as part of an erg system (aeolian dune desert), covering more than 16,000 km$^2$ on the eastern side of the Iberian Peninsula [18, 59].

According to [18, 59], this erg system (Fig 2B) displayed a threefold zonation developed along a NW–SE direction characterised by: (i) a back-erg area close to the main source area of siliciclastics (Iberian Massif), dominated by coarser-grained wadi sediments and minor aeolian accumulations; (ii) a central-erg area corresponding to the main locus of aeolian dune accumulation; and (iii) a fore-erg area characterised by the interplay between aeolian dunes and shallow marine deposits from the Western Tethys, giving rise to a complex interaction of tidally-reworked aeolian dune and subtidal facies associations.

In the Maestrazgo Basin, the Utrillas Group is overlain by the Mosqueruela Formation [50, 57, 59, 63] (Fig 2B), a marine carbonate unit, that represents the onset of carbonate sedimentation (Southern Iberian Ramp), which will prevail during most of the Late Cretaceous [48, 64]. The Maestrazgo Basin is divided into several sub-basins (Fig 1) that include: the Galve, Oliete, Las Parras, Morella, El Perelló, La Salzedella, and Peñagolosa sub-basins [49].

In this paper, three stratigraphic sections have been measured in three outcrops in the Maestrazgo Basin, namely: San Just (according to [31]: "left slope of San Just"), Arroyo de la Pascueta and Cortes de Arenoso (according to [40]: "La Hoya outcrop") sections. The San Just section is located in the Las Parras Sub-basin, about 70 km to the NNW from the other studied sections which crop out in the NNW sector of the Penyagolosa Sub-basin (Fig 1). The studied sedimentary succession is located in the distal part of the mid-Cretaceous desert system, which is characterised by a fore-erg setting where aeolian strata interacted with shallow marine deposits in the coastal fringe of the Western Tethys.

**San Just section (SJ).** The 22 m-thick section is located on a roadside (N-420 road), approximately 3 km to the south of Utrillas village (Teruel Province) and overlain by the marine carbonates of the Mosqueruela Formation, dated as Cenomanian [46]. The succession can be divided into a lower part mainly consisting of dark siltstones and mudstones interspersed with cm- to dm-thick fine-grained sandstone layers with wavy to flaser heterolithic laminations (Fig 3), and an upper part dominated by fine-grained, cross-bedded, sandstones with thin (dm-thick) interbedded mudstones, which may include water deformation structures and slumps. The amber-bearing bed is located in the lower part of the section and is overlain by (large-scale) planar cross-bedded fine-grained sandstones exhibiting sharp basal contacts. Kindney-shaped amber pieces together with "stalactitic aerial" amber pieces, the latter with bioinclusions, are abundant in this bed [43] (Fig 4A and 4B).

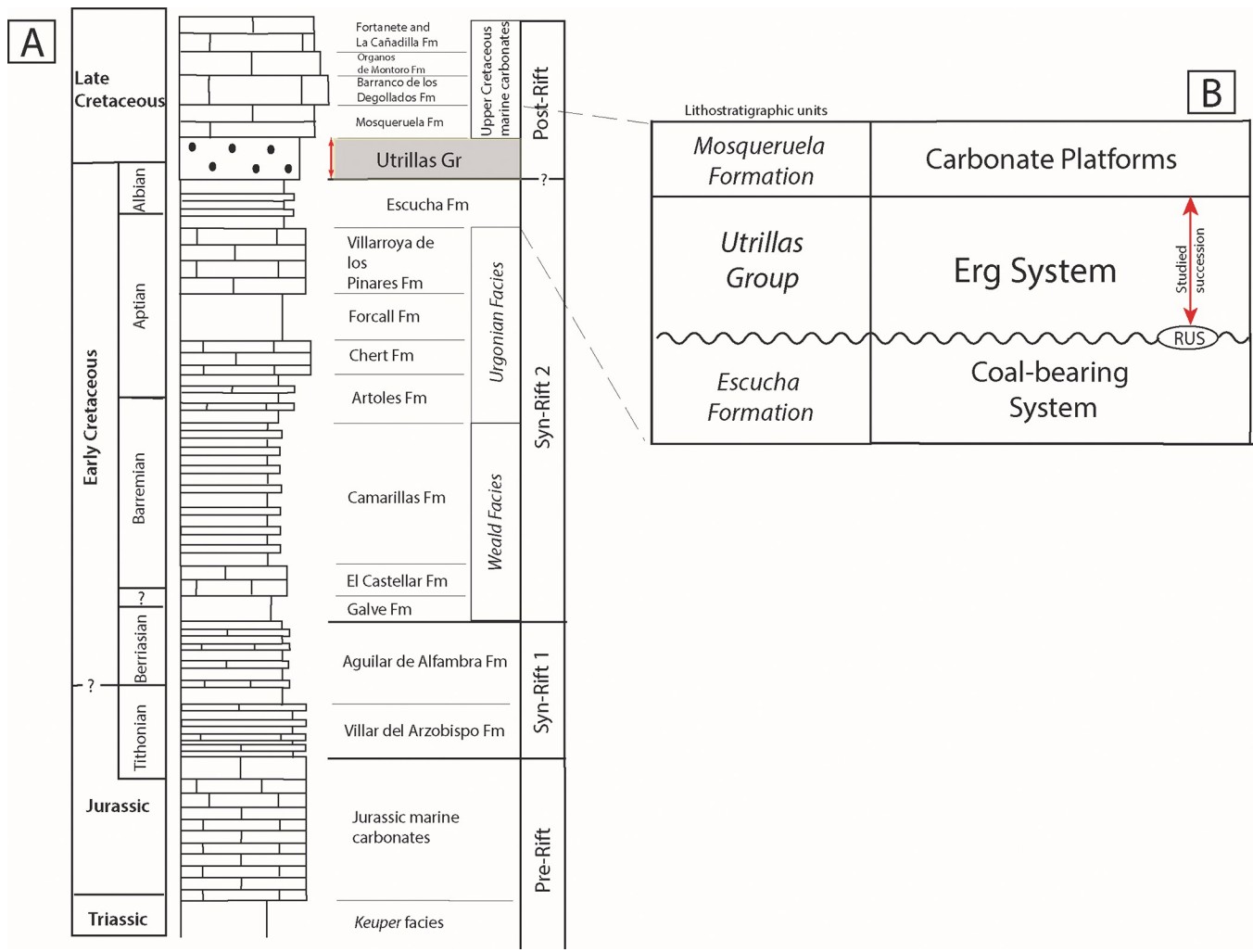

**Fig 2. Synthetic stratigraphic log of the Maestrazgo Basin from the Triassic to the Upper Cretaceous.** (A) The red arrow shows the location of the Utrillas Group in grey colour. Based on data from [52, 55]. (B) Detail of the lithostratigraphy of the studied succession, where the Utrillas Group Erg System overlies the coal-bearing deposits of the Escucha Formation with a regional unconformity surface (RUS), which marks the onset of the Erg System. The Utrillas Group succession is, in turn, overlain by marine carbonate deposits of Mosqueruela Formation. Based on data from [59].

**Cortes de Arenoso section (CA).** This 172 m-thick section is cropping out in a ravine in the NNW of the Cortes de Arenoso locality (Castellón Province). The succession, sharply overlain by marine carbonates attributed to the Mosqueruela Formation [47, 65], is dominated by fine-grained sandstones displaying large-scale planar cross-bedded sets, which laterally shift into dark mudstones and siltstones containing plant megaremains, especially abundant in the first 10–30 m of the logged section (Fig 4). The section includes heterolithic siltstone/mudstone and sansdtone deposits displaying subaqueous tidal sedimentary structures such as double mud drapes, lenticular, flaser and wavy laminations and cross-bedded sets depicting current bipolarity (herringbone-like cross-beddings). The amber-bearing level corresponds to the La Hoya amber outcrop (not to confuse it with the Barremian compression outcrop of Las Hoyas, also in Spain). Concretely, this level is located in the uppermost mudstone interval of the section and correlates with the amber-bearing stratum described in Arroyo de la Pascueta [40, 44].

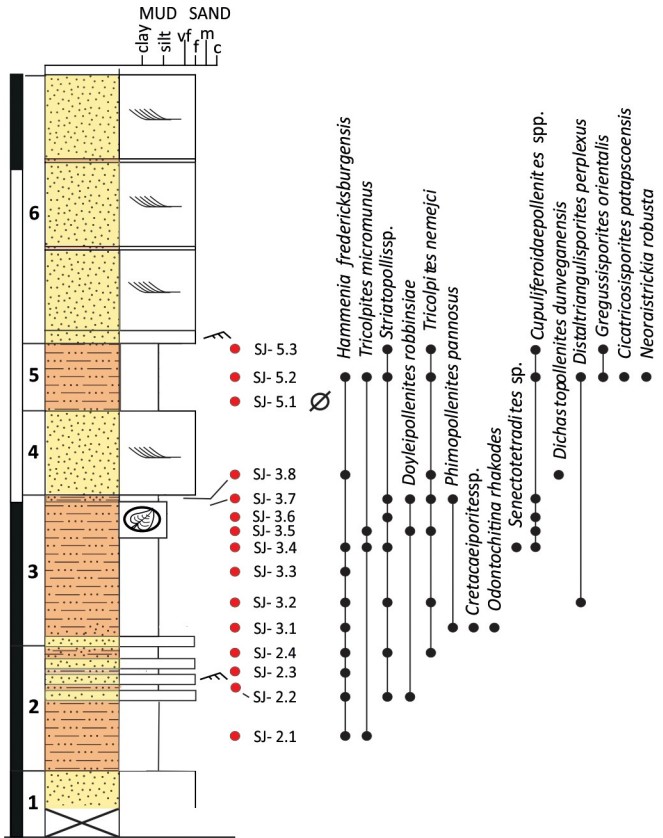

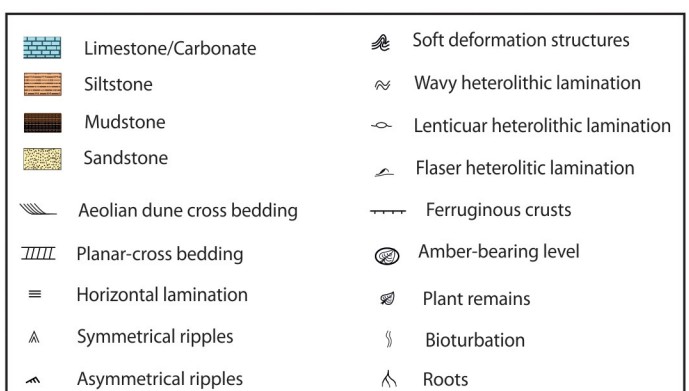

**Fig 3. Stratigraphic log of the upper Albian San Just outcrop (SJ), showing the distribution of selected biostratigraphically relevant palynomorphs.** The San Just amber outcrop is at the top of the level 3. Red dots: location of the collected samples. Black dots: stratigraphic distribution of the selected taxa. Ø: barren sample.

**Arroyo de la Pascueta section (AP).** The 182 m-thick section consists of an alternation of fine-grained homometric sandstone bodies with large-scale planar cross-bedding (Fig 6). The sandstone laterally shifts into dark siltstone to mudstone and heterolithic deposits with mud drapes and wavy to flaser laminations, showing similar characteristics to the heterolithic silt-stone/mudstone described in the Cortes de Arenoso section (Fig 5). Despite plant remains being common and evenly distributed throughout the succession, only the lower part of the section (interval between 20 to 27 m in Fig 6) was sampled and analysed in previous

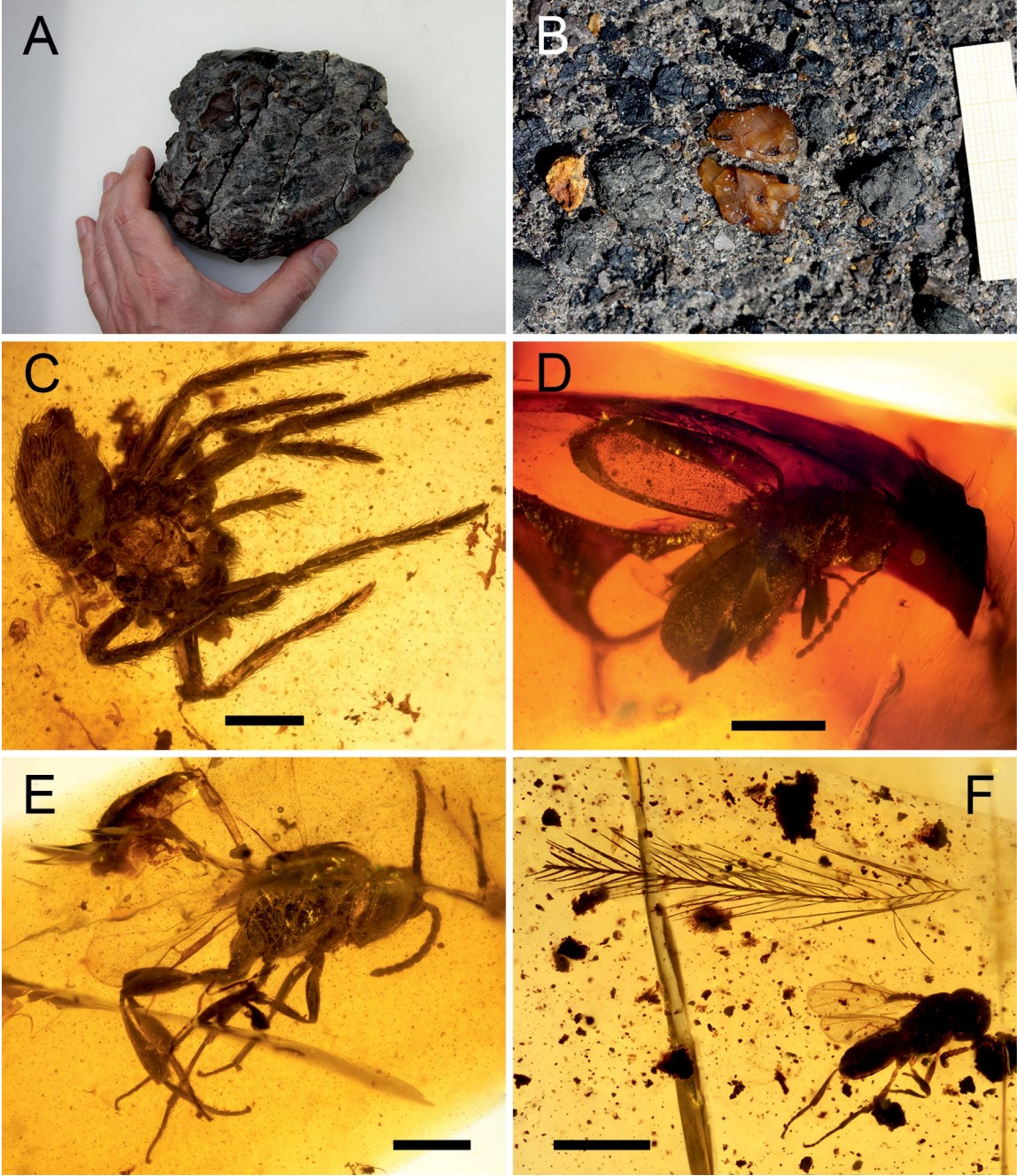

**Fig 4. Amber from the late Albian San Just outcrop (Utrillas, Teruel) and its bioinclusions.** (A) Big kidney-shaped amber mass of 640 g. (B) Unearthed amber pieces. (C) Undetermined Lagonomegopidae? spider (SJNB2012-12-07). (D) Holotype of *Actenobius magneoculus* (Coleoptera, Ptinidae) (MAP-7727). (E) Holotype of *Cretevania alcalai* (Hymenoptera, Evaniidae) (CPT-960). (F) Avian dinosaur feather barb close to an undetermined Platygastroidea wasp (CPT-4078 and CPT-4079, respectively). Scale bars C–F, 0.5 mm. Images A and B are provided by Enrique Peñalver. Acronyms SJNB, CPT and MAP correspond to amber pieces housed at the Museo Aragonés de Paleontología (Fundación Conjunto Paleontológico de Teruel-Dinópolis) in Teruel, Spain.

palaeobotanical studies [29, 30]. Amber fragments are abundant [40, 44] and recorded in a mudstone bed of the upper part of the succession, which is sharply overlain by the marine carbonates of the Mosqueruela Formation [47].

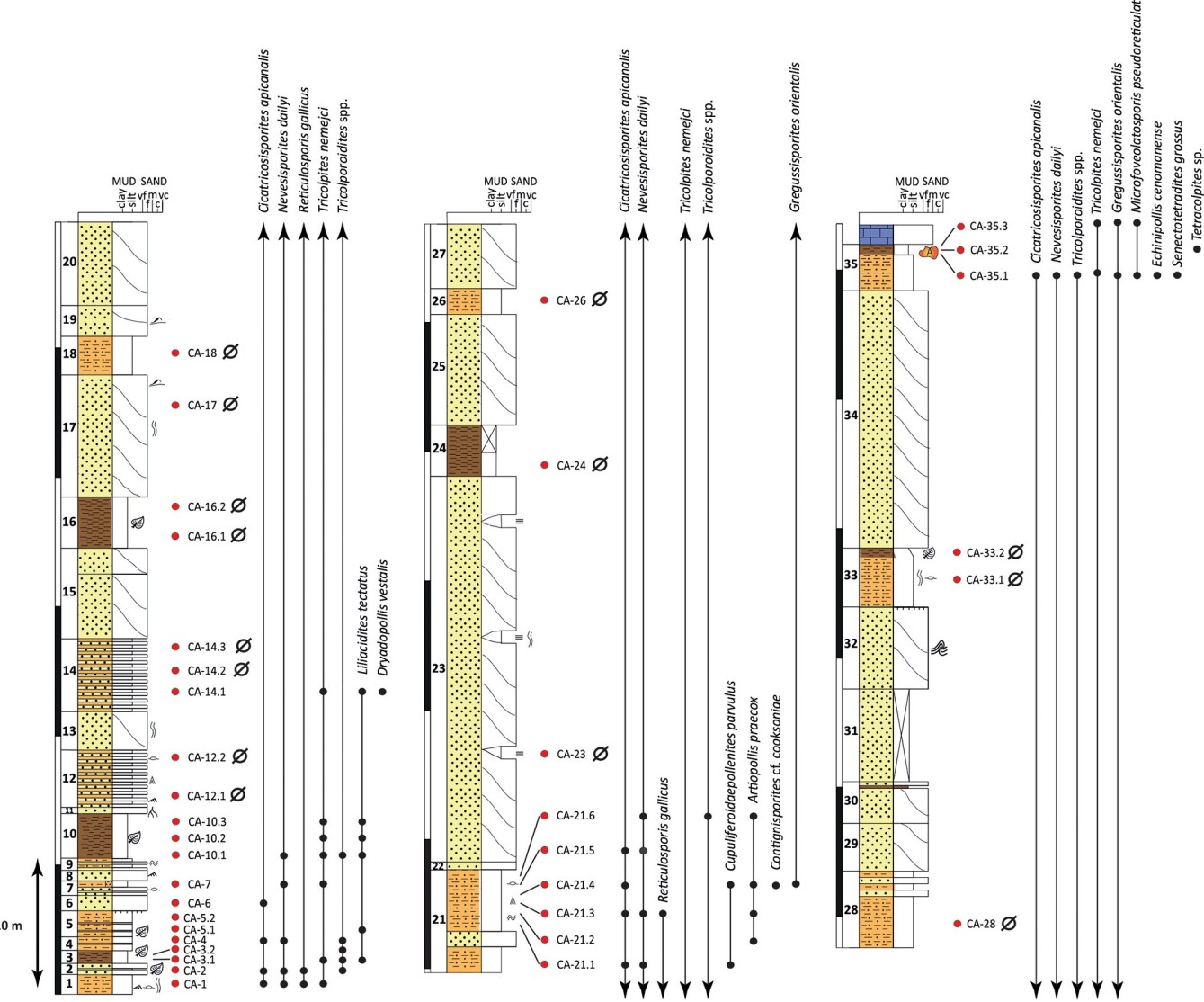

**Fig 5. Stratigraphic log of the upper Albian/lower Cenomanian Cortes de Arenoso section (CA), showing the distribution of selected biostratigraphically relevant miospores (see the legend in Fig 3).** The La Hoya amber outcrop is at the top of the section (level 35). Red dots: location of the collected samples. Black dots: stratigraphic distribution of the selected taxa. Ø: barren samples.

## Palynology

Ninety-two samples have been collected in the three cropping out successions described above (Fig 1). Although sampling primarily focused on amber-bearing strata, all suitable levels of the sections were carefully sampled (Figs 3–5) in order to obtain a more accurate and continuous biostratigraphic framework. The three logged sections were drawn using the software SedLog [66] in combination with Adobe Illustrator (www.adobe.com).

A total of 15 rock samples from the San Just, 36 from the Cortes de Arenoso and 41 from the Arroyo de la Pascueta sections were prepared for palynological analysis. Palynological residues were processed using acid digestion with HCl and HF at high temperatures [67, 68]. When required, a short oxidation with $HNO_3$ ("nitric wash") was performed in some residues. The residues were then concentrated by sieving through 500, 250 and 10 μm sieves, mounted

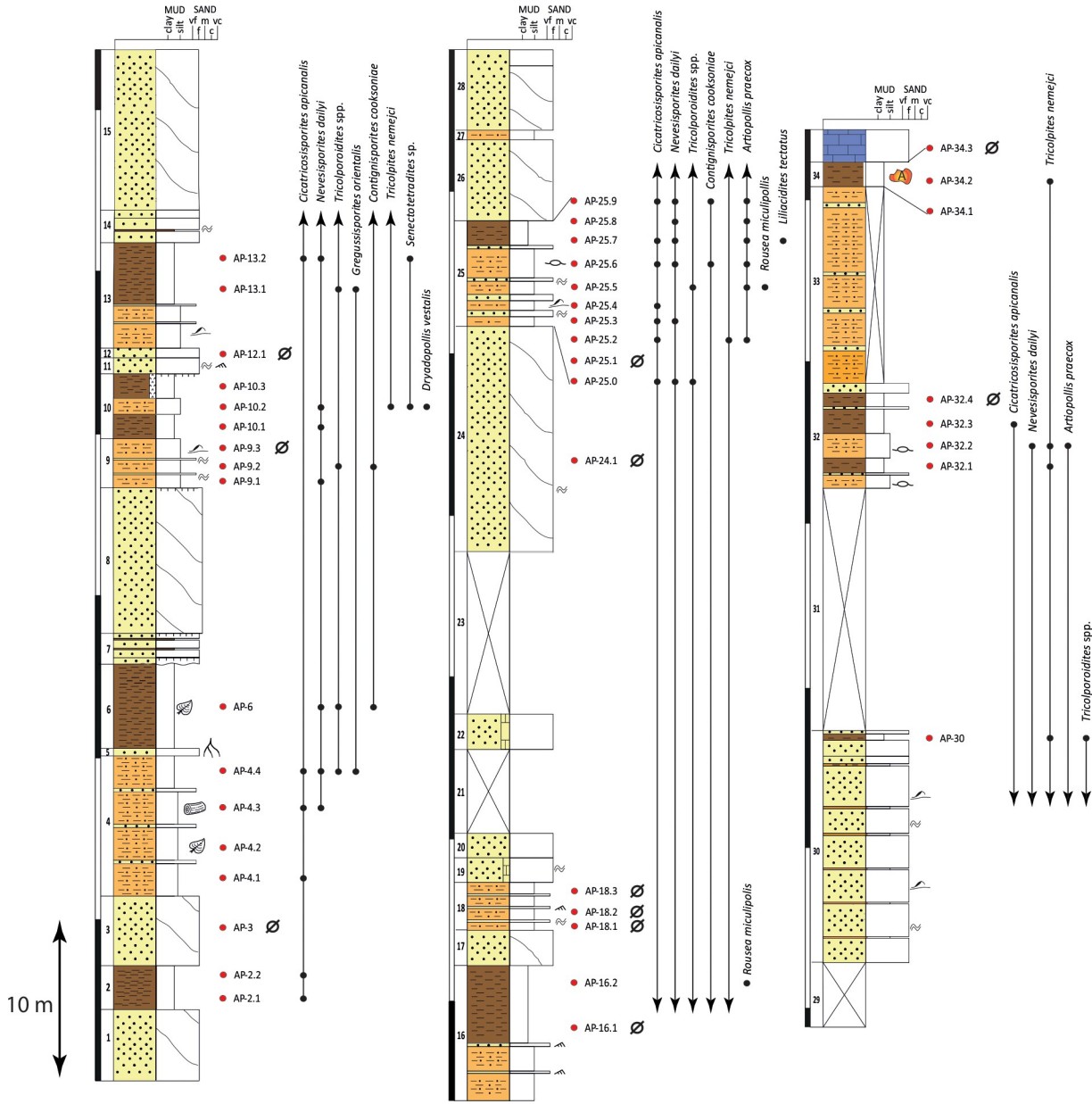

**Fig 6. Stratigraphic log of the upper Albian Arroyo de la Pascueta section (AP), showing the distribution of selected biostratigraphically relevant miospores (see the legend in Fig 3).** The Arroyo de la Pascueta amber outcrop is at the top of the section (level 34). Red dots: location of the collected samples. Black dots: stratigraphic distribution of the selected taxa. Ø: barren samples.

in glycerin jelly on strew slides. In order to include rare taxa with potential biostratigraphic value, between 500 and 1000 palynomorphs were identified per sample (S1 Appendix). Microscopic analysis of the palynological slides was performed with an Olympus BX51 microscope, incorporating a ColorView IIIu camera using a 100X oil immersion objective. Pollen diagrams were constructed by using Tilia/TGView 2.0.2 softwares [69, 70]. All studied slides are provisionally stored in the museum of the Geological Survey of Spain (CN IGME-CSIC, Madrid). A list of the identified taxa along with their botanical affinity and occurrences in each section is given in Table 1 and S1 and S2 Appendices.

**Table 1. List ordered alphabetically of the palaeobotanical taxa considering their botanical affinities, and possible preferred environment which are based in bibliographic references for aquatic palynomorphs [71, 72], ferns and allied [73–76], gymnosperms [12, 24, 74, 77–91] and angiosperms [12, 92–99].**

| Taxon | Section | Known or probable parent affinity | Possible preferred environment |
|---|---|---|---|
| *Abietinaepollenites* sp. | SJ | Gymnospermophyta; uncertain affinity | Canopy/xerophytic (upland, well drained) |
| *Acritosporites* cf. *kyrtomus* | CA | Monilophyta; uncertain affinities (Lygodiaceae??) | Ground cover/hygrophytic? |
| *Acritosporites* sp. | SJ | | |
| *Aequitriradites spinulosus* | CA | Hepaticophyta | Ground cover/xerophytic? |
| *Aequitriradites* sp. | AP | | |
| *Afropollis jardinus* | SJ, CA, AP | Gymnospermophyta; uncertain affinity | Canopy?, shrub?/xerophytic (lowland?) |
| *Antulsporites clavus* | SJ | Bryophyta; Sphagnales; Sphagnaceae | Ground cover/hygrophytic (marshes, peat bog?) |
| cf. *Antulsporites* sp. | AP | Bryophyta?; Sphagnales? | Ground cover/hygrophytic (marshes, peat bog?) |
| *Alisporites bilateralis* | CA, AP | Gymnospermophyta; uncertain affinity (Corystospermales?, Podocarpaceae?) | Canopy/xerophytic (coastal?, upland?) |
| *Alisporites grandis* | CA, AP | | |
| *Alisporites* spp. | CA | | |
| *Appendicisporites bifurcatus* | AP | Monilophyta; Schizaeales; Anemiaceae | Ground cover, understory, epiphytes/ hygrophytic (lowland, marshes, wet environments) |
| *Appendicisporites bilateralis* | SJ, CA, AP | | |
| *Appendicisporites* cf. *concentricus* | CA, AP | | |
| *Appendicisporites* cf. *crenimurus* | CA | | |
| *Appendicisporites cristatus* | CA | | |
| *Appendicisporites erdtmanii* | CA, AP | | |
| *Appendicisporites fucosus* | CA, AP | | |
| *Appendicisporites* cf. *fucosus* | SJ | | |
| *Appendicisporites* cf. *jansonii* | CA, AP | | |
| *Appendicisporites potomacensis* | CA | | |
| *Appendicisporites* cf. *potomacensis* | AP | | |
| *Appendicisporites problematicus* | AP | | |
| *Appendicisporites* cf. *problematicus* | CA, AP | | |
| *Appendicisporites robustus* | CA | | |
| *Appendicisporites tricornitatus* | CA, AP | | |
| *Appendicisporites* cf. *tricornitatus* | CA | | |
| *Appendicisporites* spp. | SJ, CA, AP | | |
| ?*Apteodinium* sp. | CA | Dinoflagellata: Gonyaulacaceae | Marine |
| *Araucariacites australis* | SJ, CA, AP | Gymnospermophyta; Coniferales; Araucariaceae | Canopy/xerophytic (coastal, upland?) |
| *Artiopollis praecox* | CA, AP | Antophyta; non-magnoliid dicotyledon | Ground cover?/hygrophytic? |
| *Asteropollis asteroides* | CA | Antophyta; Chloranthaceae | Ground cover/hygrophytic (river banks, pioneer) |
| *Baculatisporites comaumensis* | SJ, CA, AP | Monilophyta; Osmundaceae | Ground cover/hygrophytic (riparian, mires, marshes) |
| *Baculatisporites* spp. | CA, AP | | |
| *Balmeiopsis limbata* | SJ, CA, AP | Gymnospermophyta; Coniferales; Araucariaceae | Canopy/xerophytic (coastal, upland?) |
| *Biretisporites potoniaei* | CA, AP | Monilophyta; uncertain affinities | Ground cover/hygrophytic (understory, wet lowland) |
| *Biretisporites* spp. | SJ, CA | | |
| *Botryococcus braunii* | SJ, CA, AP | Chlorophyta; Botryococcaceae | Brackish environments, fresh water bogs, temporary pools, ponds and lakes |
| *Brachyphyllum* cf. *obesum* | SJ | Gymnospermophyta; Coniferales; Araucariaceae | Canopy/xerophytic (coastal) |
| ?*Callaiosphaeridium assymetricum* | AP | Dinoflagellata: Gonyaulacaceae | Marine |

*(Continued)*

**Table 1.** (Continued)

| Taxon | Section | Known or probable parent affinity | Possible preferred environment |
|---|---|---|---|
| *Callialasporites dampieri* | SJ, CA, AP | Gymnospermophyta; Coniferales; Araucariaceae/Podocarpaceae | Canopy/xerophytic (coastal, upland?) |
| *Callialasporites minor* | CA | | |
| *Callialasporites trilobatus* | CA, AP | | |
| *Callialasporites turbatus* | CA | | |
| *Callialasporites segmentatus* | AP | | |
| *Camarozonosporites ambigens* | CA | Lycophyta; Lycopodiaceae | Ground cover/hygrophytic (lowland, mires, wet environments) |
| *Camarozonosporites* cf. *ambigens* | CA | | |
| *Camarozonosporites* cf. *hammenii* | CA | | |
| *Camarozonosporites insignis* | AP | | |
| *Camarozonosporites* spp. | CA, AP | | |
| *Cardioangulina* sp. | CA | Monilophyta; uncertain affinities | Ground cover/hygrophytic? |
| *Cedripites* cf. *canadensis* | SJ | Gymnospermophyta; Coniferales; Pinaceae | Canopy/xerophytic (upland, well drained) |
| *Cedripites* cf. *cretaceous* | CA | | |
| *Cedripites* sp. | CA, AP | | |
| *Ceratosporites* sp. | SJ, CA | Lycophyta; Selaginellaceae | Ground cover/hygrophytic? |
| *Cerebropollenites macroverrucosus* | CA, AP | Gymnospermophyta; uncertain affinity (taxodioid, Sciadopityaceae) | Canopy/hygrophytic? (pioneer?) |
| ?*Chichaouadinium arabicum* | SJ | Dinoflagellata: Peridiniaceae | Marine |
| *Chlamydophorella* sp. | CA | Dinoflagellata: uncertain affinity | Marine |
| *Chomotriletes minor* | SJ, CA, AP | Chlorophyta; Zygnemataceae | Freshwaters |
| *Cibotiumspora jurienensis* | CA, AP | Monilophyta; uncertain affinities | Ground cover/hygrophytic (understory, wet lowland) |
| *Cibotiumspora* sp. | SJ | | |
| *Cicatricosisporites angicanalis* | CA | Monilophyta; Schizaeales; Anemiaceae | Ground cover, understory, epiphytes/ hygrophytic (lowland, marshes, wet environments) |
| *Cicatricosisporites apicanalis* | SJ, CA, AP | | |
| *Cicatricosisporites aralicus* | SJ, AP | | |
| *Cicatricosisporites crassiterminatus* | CA | | |
| *Cicatricosisporites dorogensis* | CA | | |
| *Cicatricosisporites* cf. *grabowensis* | SJ | | |
| *Cicatricosisporites hughesii* | SJ, CA, AP | | |
| *Cicatricosisporites* cf. *ludbrookiae* | AP | | |
| *Cicatricosisporites* cf. *minutaestriatus* | AP | | |
| *Cicatricosisporites patapscoensis* | SJ, CA, AP | | |
| *Cicatricosisporites potomacensis* | SJ, CA, AP | | |
| *Cicatricosisporites proxiradiatus* | CA | | |
| *Cicatricosisporites pseudotripartitus* | CA | | |
| *Cicatricosisporites* cf. *pseudotripartitus* | SJ, CA, AP | | |
| *Cicatricosisporites* cf. *recticicatricosus* | CA | | |
| *Cicatricosisporites sinuosus* | CA, AP | | |
| *Cicatricosisporites sprumontii* | CA | | |
| *Cicatricosisporites venustus* | SJ, CA, AP | | |
| *Cicatricosisporites* spp. | SJ, CA, AP | | |

(*Continued*)

**Table 1.** (Continued)

| Taxon | Section | Known or probable parent affinity | Possible preferred environment |
|---|---|---|---|
| *Cicatricosospoites auritus* | CA | Monilophyta; Schizaeales; Anemiaceae | Ground cover, understory, epiphytes/ hygrophytic (lowland, marshes, wet environments) |
| *Cicatricososporites phaseolus* | CA | | |
| *Cingutriletes clavus* | CA | Bryophyta; Sphagnales; Sphagnaceae | Ground cover/hygrophytic (marshes, peat bog?) |
| *Cingutriletes congruens* | SJ, CA | | |
| *Cingutriletes* sp. | CA, AP | | |
| *Circulodinium brevispinosum* | CA | Dinoflagellata: Areoligeraceae | Marine |
| *Classopollis major* | SJ, CA, AP | Gymnospermophyta; Coniferales; Cheirolepidiaceae | Canopy or shrubs/xerophytic (coastal, well drained lowland) |
| *Classopollis* cf. *obidosensis* | SJ, CA, AP | | |
| *Classopollis* spp. | SJ, CA, AP | | |
| *Clavatipollenites hughesii* | SJ, CA, AP | Antophyta; Chloranthaceae | Ground cover/hygrophytic (river banks, pioneer) |
| *Clavatipollenites tenellis* | SJ, CA, AP | | |
| *Clavatipollenites* cf. *tenellis* | CA | | |
| *Clavatipollenites* spp. | SJ, CA, AP | | |
| *Clavifera triplex* | CA | Monilophyta; Gleicheniaceae | Ground cover/xerophytic (lowland, pioneer) |
| *Clavifera* sp. | SJ, CA | | |
| *Cometodinium* sp. | CA | Dinoflagellata: Gonyaulacaceae | Marine |
| *Concavissimisporites* cf. *crassatus* | SJ, CA, AP | Monilophyta; Dicksoniaceae/Lygodiaceae | Ground cover/hygrophytic (understory, wet lowland) |
| *Concavissimisporites granulatus* | CA, AP | | |
| *Concavissimisporites* cf. *irregularis* | SJ, AP | | |
| *Concavissimisporites punctatus* | CA | | |
| *Concavissimisporites verrucosus* | CA | | |
| *Concavissimisporites* spp. | SJ, CA, AP | | |
| *Contignisporites cooksoniae* | SJ, AP | Monilophyta; Pteridaceae | Ground cover/hygrophytic (understory, wet lowland) |
| *Contignisporites* cf. *cooksoniae* | CA | | |
| *Contignisporites* sp. | CA, AP | | |
| *Converrucosisporites geniculatus* | SJ, AP | Monilophyta; uncertain affinities | Ground cover/hygrophytic? |
| *Converrucosisporites* spp. | CA, AP | | |
| *Costatoperforosporites fistulosus* | CA, AP | Monilophyta; Schizaeales; Anemiaceae | Ground cover, understory, epiphytes/ hygrophytic (lowland, marshes, wet environments) |
| *Costatoperforosporites foveolatus* | SJ, CA, AP | | |
| *Costatoperforosporites triangulatus* | CA, AP | | |
| *Costatoperforosporites* cf. *triangulatus* | SJ | | |
| *Costatoperforosporites* spp. | SJ, CA, AP | | |
| *Couperisporites* sp. | AP | Hepaticophyta | Ground cover/xerophytic? |
| *Crassipollis chaloneri* | SJ, CA, AP | Antophyta; uncertain affinity | Ground cover?/hygrophytic? |
| *Crassipollis* sp. | CA | | |
| *Cretacaeiporites* spp. | SJ, CA, AP | Antophyta; uncertain affinity (Trimeniaceae, Ranunculaceae, Eudicots) | Ground cover?/hygrophytic? |
| *Cribroperidinium* spp. | AP | Dinoflagellata: Gonyaulacaceae | Marine |

*(Continued)*

**Table 1.** (Continued)

| Taxon | Section | Known or probable parent affinity | Possible preferred environment |
|---|---|---|---|
| *Crybelosporites pannuceus* | SJ, CA, AP | Monilophyta; Marsileaceae | Ground cover/hygrophytic |
| *Cupuliferoidaepollenites parvulus* | CA | Antophyta; non-magnoliid dicotyledon | Canopy?/xerophytic? (lowland) |
| *Cupuliferoidaepollenites* sp. | SJ, CA, AP | | |
| *Cyathidites australis* | SJ, CA, AP | Monilophyta; uncertain affinities | Ground cover/hygrophytic (understory, wet lowland) |
| *Cyathidites minor* | SJ, CA, AP | | |
| *Cycadopites carpentieri* | CA | Gymnospermophyta; uncertain affinity (Bennettitales, Cycadales, Czekanowskiales, Ginkgoales, Gnetales, Pentoxylales) | Canopy/xerophytic (lowland, coastal) |
| *Cycadopites follicularis* | SJ, CA, AP | | |
| *Cycadopites* spp. | SJ, CA, AP | | |
| *Cyclonephelium chabacca* | CA, AP | Dinoflagellata: Areoligeraceae | Marine |
| *Cyclonephelium vannophorum* | CA | | |
| *Cyclonephelium* spp. | SJ, CA, AP | | |
| *Cymathiosphaera* spp. | CA, AP | Chlorophyta (Prasinophyte) | Marine |
| *Dammarites* cf. *albens* | AP | Gymnospermophyta; Coniferales; Araucariaceae | Canopy/xerophytic (coastal) |
| *Dammarites* spp. | SJ | | |
| *Dapsilidinium* sp. | SJ | Dinoflagellata: uncertain affinity | Marine |
| *Deltoidospora* spp. | SJ, CA, AP | Monilophyta; uncertain affinities | Ground cover/hygrophytic (understory, wet lowland) |
| *Densoisporites velatus* | CA | Lycophyta; Pleuromeiaceae? | Ground cover/hygrophytic (coastal, mires) |
| *Densoisporites* sp. | SJ, AP | | |
| *Dichastopollenites dunveganensis* | SJ, CA | Antophyta; uncertain affinity | Ground cover?, floating plant?/hygrophytic |
| *Dichastopollenites ghazalatensis* | CA | | |
| *Dichastopollenites* cf. *reticulatus* | AP | | |
| *Dichastopollenites* spp. | SJ, CA, AP | | |
| *Dictyophyllidites equiexinus* | CA, AP | Monilophyta; Dipteridaceae/Matoniaceae | Ground cover/hygrophytic? |
| *Dictyophyllidites harrisii* | SJ, CA, AP | | |
| *Dictyophyllidites* sp. | CA | | |
| *Distachyapites* sp. | SJ | Gymnospermophyta; Gnetales; Ephedraceae | Shrubs?/xerophytic (lowland, coastal, upland) |
| *Distaltriangulisporites irregularis* | CA | Monilophyta; Schizaeales; Schizaeaceae | Ground cover/hygrophytic (understory, wet lowland) |
| *Distaltriangulisporites maximus* | CA | | |
| *Distaltriangulisporites* cf. *maximus* | CA | | |
| *Distaltriangulisporites perplexus* | SJ | | |
| *Distaltriangulisporites* cf. *perplexus* | CA | | |
| *Distaltriangulisporites* sp. | CA | | |
| *Distaverrusporites* sp. | AP | Monilophyta; uncertain affinities | Ground cover/hygrophytic? |
| *Doyleipollenites robbinsiae* | SJ, AP | Antophyta; uncertain affinity | Ground cover?/hygrophytic? |
| *Dryadopollis vestalis* | CA, AP | Antophyta; non-magnoliid dicotyledon | Ground cover?, shrubs?/xerophytic? (lowland) |
| *Echimonocolpites* sp. | AP | Antophyta; uncertain affinity (Nymphaeaceae?, Monocots?) | Floating plants/hygrohytic (mire, marshes) |
| *Echinatisporis varispinosus* | SJ | Lycophyta; Selaginellaceae | Ground cover, xerophytic? |
| *Echinatisporis* sp. | CA, AP | | |
| *Echinipollis cenomanensis* | CA | Antophyta; uncertain affinity | Ground cover?/xerophytic? |

(*Continued*)

**Table 1.** (*Continued*)

| Taxon | Section | Known or probable parent affinity | Possible preferred environment |
|---|---|---|---|
| *Equisetosporites barghoornii* | CA | Gymnospermophyta; Gnetales; Ephedraceae | Shrub?/xerophytic (lowland, coastal, upland) |
| *Equisetosporites multicostatus* | CA, AP | | |
| *Equisetosporites* aff. *rousei* | SJ | | |
| *Equisetosporites* spp. | SJ, CA, AP | | |
| *Eretmophyllum* sp. | SJ | Gymnospermophyta; Ginkgoales | Canopy/xerophytic (lowland, marshes) |
| *Eucommiidites minor* | SJ, CA, AP | Gymnospermophyta; Erdtmanithecales | Canopy or shrub/xerophytic (lowland) |
| *Eucommiidites troedsonii* | SJ, CA, AP | | |
| *Eucommiidites* spp. | CA | | |
| *Exesipollenites tumulus* | SJ, CA, AP | Gymnospermophyta; uncertain affinity (Bennettitales, taxodioid conifers) | Canopy or shrub/xerophytic (coastal?, lowland?) |
| *Exesipollenites* sp. | SJ, CA, AP | | |
| *Exochosphaeridium* sp. | AP | Dinoflagellata: uncertain affinity | Marine |
| *Florentinia mantellii* | SJ, AP | Dinoflagellata: Gonyaulacaceae | Marine |
| *Florentinia* sp. | CA | | |
| *Foraminisporis asymmetricus* | SJ, CA, AP | Hepaticophyta | Ground cover, hygrophytic? |
| *Foraminisporis* cf. *undulatus* | CA | | |
| *Foraminisporites* sp. | CA | | |
| *Foveosporites* cf. *subtriangularis* | CA, AP | Lycophyta | Ground cover/hygrophytic (lowland, mires, wet environments) |
| *Foveotricolpites concinnus* | CA | Antophyta; non-magnoliid dicotyledon | Canopy?/xerophytic? (lowland) |
| *Foveotricolpites* spp. | CA, AP | | |
| *Foveotriletes* cf. *parviretus* | CA, AP | Monilophyta; uncertain affinities | Ground cover/hygrophytic? |
| *Fraxinoipollenites constrictus* | AP | Antophyta; eudicots? | Canopy?/xerophytic? (lowland) |
| *Fraxinoipollenites*? sp. A sensu Barrón et al. 2015 | SJ, CA, AP | | |
| *Fraxinoipollenites*? sp. B sensu Barrón et al., 2015 | SJ | | |
| *Fraxinoipollenites*? sp. | AP | | |
| *Fraxinoipollenites* spp. | CA, AP | | |
| *Frenelopsis justae* | SJ | Gymnospermophyta; Coniferales; Cheirolepidiaceae | Canopy or shrubs/xerophytic (coastal, well drained lowland) |
| *Frenelopsis turolensis* | AP | | |
| *Frenelopsis* cf. *turolensis* | CA | | |
| *Ginginodinium* cf. *evittii* | SJ | Dinoflagellata: Peridinaceae | Marine |
| *Gleicheniidites* cf. *bolchovitinae* | SJ | Monilophyta; Gleicheniaceae | Ground cover/xerophytic (lowland, pioneer) |
| *Gleicheniidites carinatus* | CA | | |
| *Gleicheniidites laetus* | SJ | | |
| *Gleicheniidites senonicus* | SJ, CA, AP | | |
| *Gleicheniidites umbonatus* | CA | | |
| *Gnetaceaepollenites oreadis* | AP | Gymnospermophyta; Gnetales; Ephedraceae | Shrub/xerophytic (lowland, coastal, upland) |
| *Granulatisporites infirmus* | CA | Monilophyta; Botryopteridales | Ground cover/hygrophytic (wet lowland) |
| *Granulatisporites michinus* | CA | | |
| *Granulatisporites* sp. | CA, AP | | |
| *Gregussisporites orientalis* | SJ, CA, AP | Monilophyta; uncertain affinities | Ground cover/hygrophytic? |

(*Continued*)

**Table 1.** (Continued)

| Taxon | Section | Known or probable parent affinity | Possible preferred environment |
|---|---|---|---|
| *Hammenia fredericksburgensis* | SJ, CA, AP | Antophyta; Chloranthaceae | Ground cover/hygrophytic (river banks, pioneer) |
| *Impardecispora* cf. *apiverrucata* | CA | Monilophyta; Lygodiaceae | Ground cover/hygrophytic (understory, wet lowland) |
| *Impletosphaeridium* spp. | SJ, CA, AP | Dinoflagellata: uncertain affinity | Marine |
| *Inaperturopollenites dubius* | SJ, CA, AP | Gymnospermophyta; Coniferales; Cupressaceae (taxodioids) | Canopy/hygrophytic (mire, marshes, river shores, wet lowland) |
| *Inaperturopollenites* spp. | SJ, CA, AP | Gymnospermophyta; Coniferales; Araucariaceae?, Cupressaceae? | Canopy/xerophytic? (lowland, coastal?) |
| *Ischyosporites crateris* | CA | Monilophyta; Schizaeales; Schizaeaceae | Ground cover/hygrophytic (understory, wet lowland) |
| *Ischyosporites* cf. *pseudoreticulatus* | AP | | |
| *Ischyosporites* cf. *punctatus* | CA | | |
| *Ischyosporites variegatus* | CA | | |
| *Ischyosporites* sp. | SJ, CA | | |
| *Jusinghipollis* cf. *ticoensis* | SJ, CA, AP | Antophyta; uncertain affinity | Ground cover?/hygrophytic?/canopy? |
| ?*Kallosphaeridium* spp | SJ | Dinoflagellata: Gonyaulacaceae | Marine |
| *Kiokansium unituberculatum* | CA, AP | Dinoflagellata: uncertain affinity | Marine |
| *Kraeuselisporites* sp. (= *Patellasporites*? *echinatus*) | SJ | Lycophyta; Selaginellaceae? | Ground cover, xerophytic? |
| *Kraeuselisporites* spp. | CA, AP | | |
| *Laevigatosporites ovatus* | CA, AP | Monilophyta; uncertain affinities | Ground cover/hygrophytic (lowland) |
| *Laevigatosporites* spp. | SJ, CA | | |
| *Leptolepidites* cf. *crassibalteus* | CA | Unknown—Lycophyta?, Monilophyta? | Ground cover/hygrophytic? |
| *Leptolepidites proxigranulatus* | CA | | |
| cf. *Leptolepidites* sp. | SJ | | |
| *Leptolepidites* spp. | SJ, CA, AP | | |
| *Liliacidites* cf. *clavatus* | CA, AP | Antophyta; probable monocot | Ground cover/hygrophytic (wet lowland, marshes) |
| *Liliacidites* cf. *kiowaensis* | CA | | |
| *Liliacidites tectatus* | CA, AP | | |
| *Liliacidites* spp. | SJ, CA, AP | | |
| "*Liliacidites*" *minutus* | SJ, CA, AP | Antophyta; Laurales | Canopy?/xerophytic? (lowland) |
| *Lophotriletes* sp. | AP | Monilophyta; Botryopteridales | Ground cover/hygrophytic (wet lowland) |
| ?*Lusatisporites dettmannae* | SJ, CA, AP | Monilophyta; uncertain affinities | Ground cover/hygrophytic? |
| *Lycopodiumsporites* sp. | AP | Lycophyta; Lycopodiaceae | Ground cover/hygrophytic (lowland, mires, wet environments) |
| cf. *Margocolporites* sp. | SJ, CA | Antophyta; Eudicots | Canopy?/xerophytic? (lowland) |
| *Matonisporites crassiangulatus* | CA | Monilophyta; Matoniaceae | Ground cover/xerophytic? (lowland) |
| *Matonisporites* sp. | AP | | |
| *Micrhystridium* sp. | SJ, CA, AP | Acritarcha: Acanthomorphitae | Marine |
| *Microfoveolatosporis pseudoreticulatus* | CA | Monilophyta; uncertain affinities | Ground cover/hygrophytic |
| *Mirovia gothanii* | CA, AP | Gymnospermophyta; Coniferales; Miroviaceae | Canopy/xerophytic (lowland, marshes) |
| *Monocolpopollenites* sp. | SJ, CA, AP | Antophyta; Arecales, Arecaceae | Canopy/xerophytic? (lowland) |

*(Continued)*

**Table 1.** (*Continued*)

| Taxon | Section | Known or probable parent affinity | Possible preferred environment |
|---|---|---|---|
| *Monosulcites minimus* | SJ, CA, AP | Gymnospermophyta; uncertain affinity (Bennettitales, Cycadales, Ginkgoales) | Canopy/xerophytic (lowland, coastal) |
| *Monosulcites* sp. 1 | CA, AP | | |
| *Monosulcites* spp. | SJ, CA, AP | | |
| *Montsechia*-type | SJ | Antophyta; *Ceratophyllum*? | Hygrophytic? |
| *Murospora* sp. | CA, AP | Monilophyta; uncertain affinities | Ground cover/hygrophytic? |
| *Nehvizdya* (*Eretmophyllum*) *penalveri* | CA, AP | Gymnospermophyta; Ginkgoales | Canopy/xerophytic (lowland, marshes) |
| *Neoraistrickia* cf. *densata* | CA | Lycophyta; Selaginellaceae | Ground cover, xerophytic? |
| *Neoraistrickia equalis* | CA | | |
| *Neoraistrickia robusta* | SJ, CA, AP | | |
| *Neoraistrickia truncata* | CA, AP | | |
| *Neoraistrickia* spp. | SJ, CA, AP | | |
| *Nevesisporites dailyi* | SJ, CA, AP | Bryophyta | Ground cover/hygrophytic (peat bog?) |
| *Nodosisporites segmentus* | AP | Monilophyta; Schizaeales; Anemiaceae | Ground cover, understory, epiphytes/hygrophytic (lowland, marshes, wet environments) |
| *Nodosisporites* sp. | CA | | |
| *Odontochitina rhakodes* | SJ | Dinoflagellata: Ceratiaceae | Marine |
| *Oligosphaeridium complex* | SJ, CA, AP | Dinoflagellata: Leptodinioideae | Marine |
| *Oligosphaeridium* spp. | SJ, CA | | |
| *Osmundacidites wellmanii* | SJ, CA, AP | Monilophyta; Osmundaceae | Ground cover/hygrophytic (riparian, mire, marsh) |
| Organic lining | SJ | Foraminifera | Marine |
| *Ornamentifera peregrina* | SJ, CA | Monilophyta; Gleicheniaceae | Ground cover/xerophytic (lowland, pioneer) |
| *Ornamentifera* cf. *tuberculata* | CA | | |
| *Ornamentifera* sp. | CA, AP | | |
| *Ovoidinium* sp. | CA | Dinoflagellata: Peridiniaceae | Marine |
| *Ovoidites* sp. | CA | Chlorophyta; Zygnemataceae | Freshwater |
| *Palaeohystrichophora* sp. | SJ | Dinoflagellata: Peridiniaceae | Marine |
| *Palaeoperidinium cretaceum* | SJ | | |
| *Parvisaccites amplus* | SJ | Gymnospermophyta; uncertain affinity (Podocarpaceae, Voltziales) | Canopy/xerophytic (upland, well drained) |
| *Parvisaccites radiatus* | CA, AP | | |
| *Parvisaccites* spp. | CA, AP | | |
| aff. *Parvisaccites* spp. | SJ | | |
| *Patellasporites tavaredensis* | SJ, CA, AP | Monilophyta; uncertain affinities | Ground cover/hygrophytic? |
| *Penetetrapites mollis* | CA | Antophyta; uncertain affinity | Canopy?/xerophytic? (lowland) |
| *Pennipollis escuchensis* | SJ, CA, AP | Antophyta; Chloranthaceae-*Ceratophyllum* clade | Ground cover/hygrophytic (wet lowland, marshes) |
| *Pennipollis* cf. *peroreticulatus* | SJ, CA, AP | | |
| *Pennipollis reticulatus* | SJ, CA, AP | | |
| *Pennipollis* spp. | SJ, CA, AP | | |

(*Continued*)

**Table 1.** (Continued)

| Taxon | Section | Known or probable parent affinity | Possible preferred environment |
|---|---|---|---|
| *Perinopollenites halonatus* | CA, AP | Gymnospermophyta; Coniferales; Cupressaceae (taxodioids) | Canopy/hygrophytic (mire, marshes, river shores, wet lowland) |
| *Peromonolites allenensis* | SJ, CA, AP | Monilophyta; uncertain affinities | Ground cover/hygrophytic? |
| *Phanerosorisporites surensis* | CA | Monilophyta; Matoniaceae | Ground cover/hygrophytic (lowland, mires, wet environments) |
| *Phanerosorisporites*? sp. | AP | | |
| *Phimopollenites* cf. *megistus* | SJ | Antophyta; dicotyledons (Tetracentraceae, Hamamelidaceae) | Canopy?/xerophytic? (lowland) |
| *Phimopollenites pannosus* | SJ | | |
| *Phimopollenites pseudocheros* | AP | | |
| cf. *Phyllocladidites inchoatus* | SJ | Gymnospermophyta; Coniferales; Podocarpaceae | Canopy/xerophytic (upland, well drained) |
| *Phyllocladidites* spp. | CA, AP | | |
| *Pinuspollenites* spp. | SJ, CA, AP | Gymnospermophyta; Coniferales; Pinaceae | Canopy/xerophytic (upland, well drained) |
| *Plicifera* sp. | CA, AP | Monilophyta; Gleicheniaceae | Ground cover/xerophytic (lowland, pioneer) |
| *Podocarpidites* spp. | SJ, CA, AP | Gymnospermophyta; Coniferales; Podocarpaceae | Canopy/xerophytic (upland, well drained) |
| *Polycingulatisporites reduncus* | CA, AP | Bryophyta; Sphagnales; Sphagnaceae | Ground cover/hygrophytic (marshes, peat bog?) |
| *Polypodiisporonites cenomanianus* | CA | Monilophyta; uncertain affinities | Ground cover/hygrophytic |
| cf. *Psilatriletes circumundulatus* | SJ | Monilophyta; uncertain affinities | Ground cover/hygrophytic |
| *Punctatisporites couperi* | CA | Monilophyta; Osmundaceae | Ground cover/hygrophytic (riparian, mire, marsh) |
| *Reticulosporis gallicus* | CA | Monilophyta; uncertain affinities | Ground cover/hygrophytic (lowland) |
| *Retimonocolpites* cf. *crassatus* | CA | Antophyta; uncertain affinity | Ground cover?/hygrophytic? |
| *Retimonocolpites dividuus* | CA, AP | | |
| *Retimonocolpites* cf. *dividuus* | SJ | | |
| *Retimonocolpites excelsus* | CA | | |
| *Retimonocolpites* cf. *Rotundus* | SJ | | |
| *Retimonocolpites* cf. *textus* | SJ | | |
| *Retimonocolpites* spp. | SJ, CA, AP | | |
| *Retitrescolpites* sp. A sensu Barrón et al., 2015 | SJ, AP | Antophyta; non-magnoliid dicotyledon | Canopy?/xerophytic? (lowland) |
| *Retitrescolpites* spp. | CA | | |
| *Retitriletes austroclavatidites* | CA, AP | Lycophyta; Lycopodiaceae | Ground cover/hygrophytic (lowland, mires, wet environments) |
| *Retitriletes clavatoides* | AP | | |
| *Retitriletes* sp. | SJ, CA, AP | | |
| *Rotverrusporites* sp. | CA | Monilophyta; Osmundaceae | Ground cover/hygrophytic (riparian, mire, marsh) |
| *Rousea georgensis* | SJ, CA, AP | Antophyta; eudicots | Canopy?/xerophytic? (lowland) |
| *Rousea* cf. *georgensis* | CA | | |
| *Rousea* cf. *delicata* | SJ | | |
| *Rousea miculipollis* | AP | | |
| *Rousea* cf. *prosimilis* | CA | | |
| *Rousea* cf. *scitula* | CA | | |
| *Rousea* spp. | CA, AP | | |
| *Ruffordiaspora* sp. | CA, AP | Monilophyta; Schizaeales; Anemiaceae | Ground cover, understory, epiphytes/hygrophytic? (lowland, marshes) |
| *Rugubivesiculites* sp. | CA | Gymnospermophyta; uncertain affinity | Canopy/xerophytic (upland, well drained) |

(*Continued*)

**Table 1.** (Continued)

| Taxon | Section | Known or probable parent affinity | Possible preferred environment |
|---|---|---|---|
| *Schizosporis reticulatus* | CA, AP | Chlorophyta (Prasinophyte) | Marine |
| *Sciadopityspollenites*? sp. | SJ, AP | Gymnospermophyta; uncertain affinity (Sciadopityaceae?) | Canopy/xerophytic (upland, well drained) |
| *Senectotetradites grossus* | CA | Antophyta; eudicots; Nelumbonaceae? | Floating plant?/hygrophytic (marshes) |
| *Senectotetradites* spp. | SJ, AP | | |
| cf. *Serialis* sp. | SJ | Antophyta; uncertain affinity | Hygrophytic? |
| *Sestrosporites pseudoalveolatus* | CA | Lycophyta; Lycopodiaceae | Ground cover/hygrophytic (lowland, mires, wet environments) |
| *Singhipollis microreticulatus* | SJ, CA | Antophyta; uncertain affinity | Ground cover?/hygrophytic? |
| *Spermatites* sp. | CA, AP | Antophyta; uncertain affinity | Hygrophytic? |
| *Spiniferites twistringiensis* | CA | Dinoflagellata: Gonyaulacaceae | Marine |
| *Spiniferites* spp. | SJ, CA | | |
| *Staplinisporites caminus* | SJ, AP | Lycophyta; Lycopodiaceae | Ground cover/hygrophytic (lowland, mires, wet environments) |
| *Staplinisporites* cf. *telatus* | AP | | |
| *Steevesipollenites* sp. | CA, AP | Gymnospermophyta; Gnetales; Ephedraceae | Shrub/xerophytic (lowland, coastal, upland) |
| *Stellatopollis* cf. *barghoornii* | SJ | Antophyta; uncertain affinity (Magnoliids, Monocots) | Groun cover/hygrophytic (pioneer, mire, marshes) |
| *Stellatopollis* sp. | AP | | |
| *Stereisporites psilatus* | SJ, CA | Bryophyta; Sphagnales; Sphagnaceae | Ground cover/hygrophytic (marshes, peat bog?) |
| *Stereisporites* spp. | SJ, CA, AP | | |
| *Striatella* cf. *balmei* | CA | Monilophyta; uncertain affinities | Ground cover/hygrophytic |
| *Striatopollis* sp. | SJ, CA, AP | Antophyta; Eudicots | Canopy?/xerophytic? (lowland) |
| *Subtilisphaera* sp. | CA | Dinoflagellata: Peridiniaceae | Marine |
| Tasmanaceae? Gen. et sp. indet. | AP | Chlorophyta (Prasinophyte) | Marine |
| *Tasmanites* sp. | CA | Chlorophyta (Prasinophyte) | Marine |
| *Taurocusporites segmentatus* | SJ, CA, AP | Bryophyta; Sphagnales; Sphagnaceae | Ground cover/hygrophytic (marshes, peat bog?) |
| *Taurocusporites* sp. | CA, AP | | |
| *Taxodiaceaepollenites hiatus* | SJ, CA, AP | Gymnospermophyta; Coniferales; Cupressaceae (taxodioids) | Canopy/hygrophytic (mire, marshes, river shores, wet lowland) |
| *Tehamadinium* sp. | CA | Dinoflagellata: Leptodinioideae | Marine |
| *Tenua hystrix* | SJ | Dinoflagellata: Gonyaulacaceae | Marine |
| *Tetracolpites* sp. | CA | Antophyta; Eudicots | Canopy?/xerophytic? (lowland) |
| *Tetraporina* sp. | SJ | Chlorophyta; Zygnemataceae | Freshwaters |
| *Tigrisporites scurrandus* | SJ | Monilophyta; Filicales | Ground cover/hygrophytic (lowland) |
| *Tigrisporites* sp. | CA | | |
| *Transitoripollis anulisulcatus* | SJ, CA, AP | Antophyta; Chloranthaceae-Ceratophyllum clade | Groun cover?/hygrophytic? |
| *Transitoripollis similis* | CA, AP | | |
| *Transitoripollis* cf. *similis* | SJ | | |
| *Transitoripollis* spp. | SJ, CA, AP | | |
| *Trichodinium castanea* | CA, AP | Dinoflagellata: Gonyaulacaceae | Marine |

*(Continued)*

**Table 1.** (Continued)

| Taxon | Section | Known or probable parent affinity | Possible preferred environment |
|---|---|---|---|
| *Tricolpites* cf. *amplifissus* | AP | Antophyta; eudicots | Canopy?/xerophytic? (lowland) |
| *Tricolpites* cf. *brnicensis* | AP | | |
| *Tricolpites* cf. *maximus* | CA, AP | | |
| *Tricolpites micromunus* | SJ, CA, AP | | |
| *Tricolpites minutus* | CA, AP | | |
| *Tricolpites* cf. *minutus* | SJ | | |
| *Tricolpites nemejci* | SJ, CA, AP | | |
| *Tricolpites* cf. *parvus* | SJ, CA, AP | | |
| *Tricolpites sagax* | CA, AP | | |
| *Tricolpites* cf. *sagax* | CA | | |
| *Tricolpites vulgaris* | AP | | |
| *Tricolpites* cf. *vulgaris* | SJ, CA, AP | | |
| *Tricolpites* sp. cf. *Retitricolpites varireticulatus* | SJ, CA | | |
| *Tricolpites* spp. | SJ, CA, AP | | |
| *Tricolporoidites* cf. *pacltovae* | CA | Antophyta; non-magnoliid dicotyledon | Canopy?/xerophytic? (lowland) |
| *Tricolporoidites* spp. | CA, AP | | |
| *Trilobosporites* spp. | SJ, CA, AP | Monilophyta; Lygodiaceae | Ground cover/hygrophytic (understory, wet lowland) |
| *Triporoletes cenomanianus* | CA | Hepaticophyta; cf. Ricciaceae | Ground cover/hydrophytic? |
| *Triporoletes radiatus* | CA | | |
| *Triporoletes reticulatus* | CA | | |
| *Triporoletes simplex* | AP | | |
| *Triporoletes* sp. | SJ, CA | | |
| *Tucanopollis crisopolensis* | SJ, CA | | |
| *Tucanopollis* spp. | CA | Antophyta; Chloranthaceae-*Ceratophyllum* clade | Groun cover?/hygrophytic? |
| *Uesuguipollenites callosus* | SJ, CA, AP | Gymnospermophyta; Coniferales; Araucariaceae | Canopy/xerophytic (coastal, upland?) |
| *Undulatisporites* cf. *pflugii* | CA | Monilophyta; uncertain affinities | Ground cover/hygrophytic |
| *Undulatisporites* cf. *rugulatus* | AP | | |
| *Undulatisporites undulapolus* | CA | | |
| *Undulatisporites* spp. | SJ, CA, AP | | |
| *Uvaesporites* sp. | CA | Lycophyta; Selaginellaceae | Ground cover/riparian?, xerophytic? |
| *Varirugosisporites tolmanensis* | CA | Monilophyta; uncertain affinities | Ground cover/hygrophytic |
| *Varirugosisporites* sp. | CA | | |
| *Verrucatosporites* sp. | CA, AP | Monilophyta; uncertain affinities | Ground cover/hygrophytic |
| *Verrucosisporites* cf. *rotundus* | CA, AP | Monilophyta; uncertain affinities | Ground cover/hygrophytic |
| aff. *Verrucosisporites kopukuensis* | CA, AP | | |
| *Verrucosisporites* spp. | CA, AP | | |
| *Virgo amiantopollis* | CA, AP | Antophyta; Eudicots (Gunnerales) | Groun cover?/hygrophytic? |
| *Vitreisporites pallidus* | SJ, CA, AP | Pteridospermatophyta; Caytoniales | Canopy/hygrophytic (lowland, deltaic, mire, floodplains) |
| *Widdringtonites* sp. | SJ | Gymnospermophyta; Coniferales; Cupressaceae | Canopy/xerophytic (lowland) |

**Table 2. List of arthropod taxa as bioinclusions identified from San Just, Arroyo de la Pascueta and La Hoya Cretaceous amber-bearing outcrops with information about autoecology of each species and possible preferred environment.**

| Arthropod taxon | Family | Order | Amber outcrop | Autoecology, preferred environment |
|---|---|---|---|---|
| *Actenobius magneoculus* | Ptinidae | Coleoptera | San Just | Wood-borer on conifers? |
| *Alavaromma orchamum* | Alavarommatidae | Hymenoptera | San Just | Parasitoid? |
| *Ametroproctus valeriae* | Ametroproctidae | Acariformes | San Just | Detritivorous, tree bark habitat |
| *Aragomantispa lacerata* | Mantispidae | Neuroptera | San Just | Predator |
| *Aragonimantis aenigma* | Incertae sedis | Mantodea | San Just | Predator |
| *Aragonitermes teruelensis* | Incertae sedis | Blattodea: Isoptera | San Just | Recycler of lignocellulose |
| *Archaeatropos alavensis* | Empheriidae | Psocodea | San Just and Arroyo de la Pascueta | Detritivorous, decayed leaf-litter habitat |
| Archaeognatha gen. et sp. indet. | | Archaeognatha | San Just | Detritivorous, decayed leaf-litter habitat |
| *Archiculicoides skalskii* | Ceratopogonidae | Diptera | San Just | Blood-feeder |
| *Arra legalovi* | Nemonychidae | Coleoptera | San Just | Pollen-feeder on Araucariaceae? |
| Blattodea gen. et sp. indet. | | Blattodea | San Just, Arroyo de la Pascueta and La Hoya | Omnivorous, decayed leaf-litter habitat |
| *Burmaphron jentilak* | Stigmaphronidae | Hymenoptera | San Just | Parasitoid? |
| *Burmazelmira grimaldii* | Archizelmiridae | Diptera | San Just | Unknown |
| Collembola gen. et sp. indet. | | Collembola | San Just | Detritivorous, decayed leaf-litter habitat |
| *Cretaceobodes martinezae* | Otocepheidae | Acariformes | San Just | Detritivorous, decayed leaf-litter habitat |
| *Cretaceomma turolensis* | Gallorommatidae | Hymenoptera | San Just | Parasitoid? |
| *Cretevania alcalai* | Evaniidae | Hymenoptera | San Just | Parasitoid |
| *Cretevania montoyai* | Evaniidae | Hymenoptera | San Just | Parasitoid |
| *Cretevania rubusensis* | Evaniidae | Hymenoptera | Arroyo de la Pascueta | Parasitoid |
| *Diameneura marveni* | Spathiopterygidae | Hymenoptera | San Just | Parasitoid? |
| Diptera gen. et sp. indet. | | Diptera | San Just, Arroyo de la Pascueta and La Hoya | Diverse autoecology |
| *Hispanothrips utrillensis* | Stenurothripidae | Thysanoptera | San Just | Herbivorous |
| *Helius turolensis* | Limoniidae | Diptera | San Just | Pollen-feeder? |
| Hemiptera gen. et sp. indet. | | Hemiptera | San Just and Arroyo de la Pascueta | Phytophagous |
| Hymenoptera gen. et sp. indet. | | Hymenoptera | San Just, Arroyo de la Pascueta and La Hoya | Diverse autoecology |
| *Hypovertex hispanicus* | Scutoverticidae | Acariformes | San Just | Detritivorous, decayed leaf-litter habitat |
| *Iberofoveopsis miguelesi* | Perforissidae | Hemiptera | San Just | Phytophagous, wet environments |
| *Leptoconops zherikhini* | Ceratopogonidae | Diptera | San Just | Blood-feeder |
| *Leptus* sp. | Erythraeidae | Acariformes | San Just | Parasitoid |
| *Litoleptis fossilis* | Rhagionidae | Diptera | San Just | Blood-feeder? |
| *Microphorites utrillensis* | Dolichopodidae | Diptera | San Just | Predator, sandy beach environment |
| *Mymaropsis turolensis* | Spathiopterygidae | Hymenoptera | San Just | Parasitoid? |
| *Orchestina* sp. | Oonopidae | Araneae | San Just | Predator |
| Orthoptera gen. et sp. indet. | Elcanidae? | Orthoptera | San Just | Phytophagous |
| *Preempheria antiqua* | Empheriidae | Psocodea | San Just | Detritivorous, decayed leaf-litter habitat |
| *Protoculicoides hispanicus* | Ceratopogonidae | Diptera | San Just | Blood-feeder |
| *Protoculicoides sanjusti* | Ceratopogonidae | Diptera | San Just | Blood-feeder |
| Pseudoscorpiones gen. et sp. indet. | | Pseudoscorpiones | San Just | Predator, decayed leaf-litter habitat |
| Psocodea gen. et sp. indet. | | Psocodea | San Just and Arroyo de la Pascueta | Detritivorous, decayed leaf-litter habitat |
| Raphidioptera gen. et sp. indet. | | Raphidioptera | Arroyo de la Pascueta | Predator |
| *Serphites silban* | Serphitidae | Hymenoptera | San Just | Parasitoid? |
| *Spinomegops aragonensis* | Lagonomegopidae | Araneae | San Just | Predator |
| *Tenuelamellarea estefaniae* | Lamellareidae | Acariformes | San Just | Detritivorous, decayed leaf-litter habitat |
| *Trhypochthonius lopezvallei* | Trhypochthoniidae | Acariformes | San Just | Detritivorous, decayed leaf-litter habitat |
| *Utrillabracon electropteron* | Braconidae | Hymenoptera | San Just | Parasitoid |

Principal Component Analysis (PCA) was performed to evaluate environmental gradients underlying the palynological dataset and to identify groups of taxa representing different types of vegetation. Only the most abundant or ecologically relevant taxa were analysed. PCA calculations were performed using the SPSS and software (version 4.0.2) with FactoMineE and Rioja packages for PCA and Cluster Analysis, respectively [100, 101].

### Mega and mesoflora

Megaflora was initially investigated and documented by using an Olympus SZX 12 stereomicroscope with DP 70 digital camera. Material with preserved morphological details was studied via Scanning Electron Microscopy (SEM). Specimens were mounted on cleaned aluminium stubs using nail polish, coated with gold and examined with a Hitachi S-3700N Environmental Scanning Electron Microscope at 2 kV, at the National Museum of Prague (Czechia), where the studied material is stored temporarily. Specimens from SJ and AP will be transferred in the future to the Museo Aragonés de Paleontología (Fundación Conjunto Paleontológico de Teruel-Dinópolis, Spain) as the definitive repository.

### Amber bioinclusions

In the Maestrazgo Basin, amber pieces with bioinclusions have been collected for research purposes since 1998. They have been prepared using epoxy resin [102], to improve the visualisation of the bioinclusions and their long-term preservation. The amber pieces from the San Just and Arroyo de la Pascueta outcrops are housed at the Museo Aragonés de Paleontología (Fundación Conjunto Paleontológico de Teruel-Dinópolis, Teruel), and the ones from the La Hoya outcrop at the Museu de la Universitat de València d'Història Natural (Valencia). Bioinclusions have been photographed using a digital camera sCMEX-20 attached to a compound microscope Olympus CX41 with the software ImageFocusAlpha version 1.3.7.12967.20180920 (www.euromex.com) and processed with Photoshop CS6 version 13.0 (www.adobe.com). The list of identified taxa is included in Table 2 and S2 Appendix.

## Results

### Palaeobotanical aspects

A total of 26 out of 92 samples processed for their palynological content were barren (1 in San Just section, 14 in Cortes de Arenoso section and 11 in the Arroyo de la Pascueta section). Productive samples showed an overall good recovery and well-preserved palynological content. Except on a few occasions, interpreted as reworking, miospores and dinocysts present a yellow to pale brown colour suggesting low thermal maturation [103]. A total of 367 palynomorph types (S2 Appendix) have been identified, mainly consisting of dinoflagellate cysts (dinocysts, 30 taxa; Fig 7C–7H), bryophyte, lycophyte and pteridophyte spores (186 taxa; Fig 8), gymnosperms (53 taxa; Fig 7I–7T) and angiosperms (88 taxa; Fig 9). A small number of acanthomorph acritarchs, phycomes of prasinophytes, freshwater algae (Fig 7A and 7B) and linings of foraminifers have also been recorded (S1 Appendix).

Unlike the upper Albian–lower Cenomanian successions of the Basque-Cantabrian and Lusitanian basins [104, 105], dinocysts were only scarcely recorded in the studied assemblages. The most frequently occurring taxa are *Cyclonephelium* spp., *Criboperidinium* spp. (Fig 7C), *Implestosphaeridium* spp., *Kiokansium unituberculatum* (Fig 7H) and *Oligosphaeridium complex* (Fig 7G). The remaining of the identified marine and freshwater palynomorphs are not numerically significant (S1 Appendix). Terrestrial assemblages are characterised by abundant conifers with *Classopollis* (mainly *Classopollis major*) and *Inaperturopollenites dubius* (Fig 7K

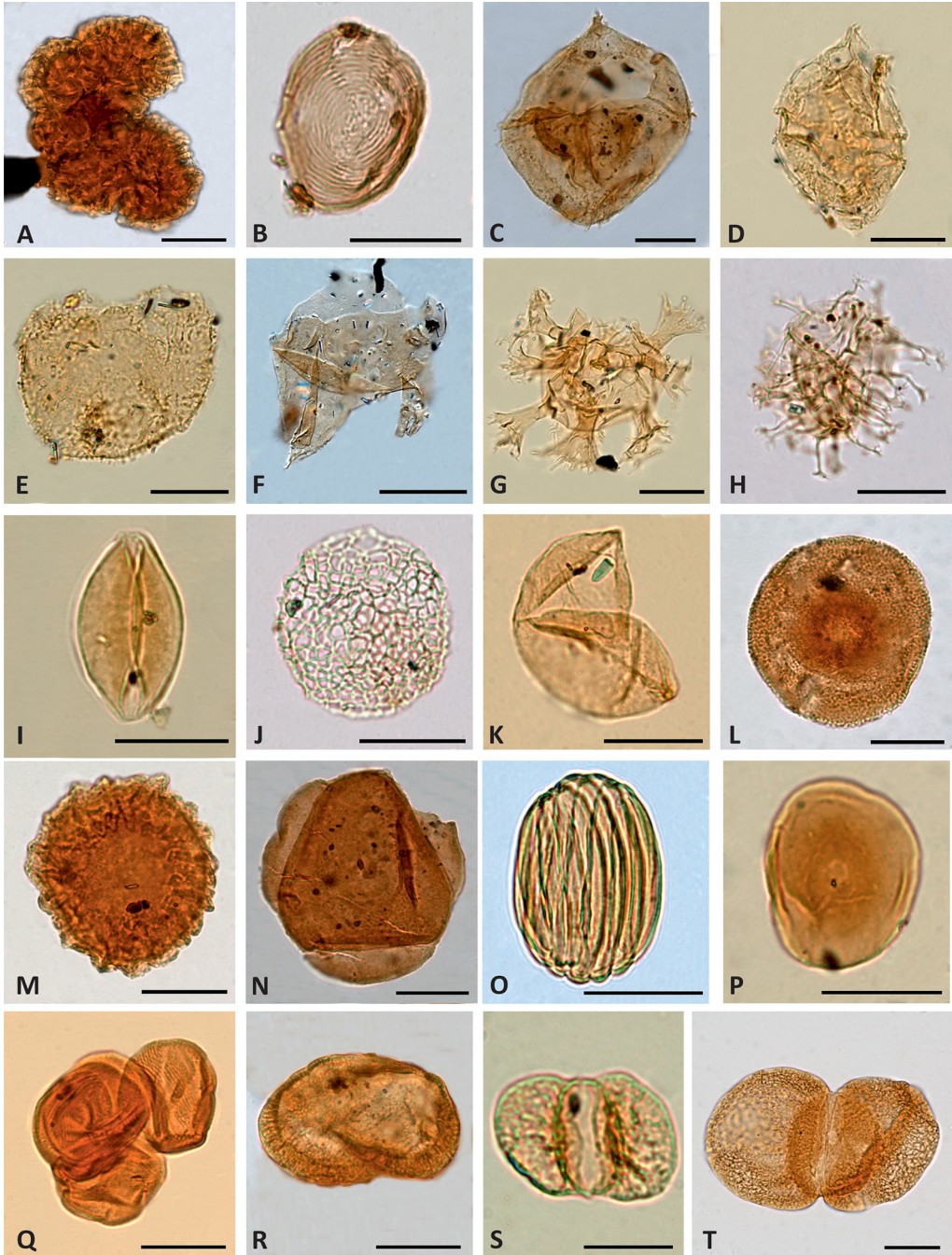

**Fig 7. Light photomicrographs of selected aquatic palynomorphs and gymnosperm pollen grains.** (A) *Botryococcus braunii*, level AP-34.1. (B) *Chomotriletes minor*, level AP-34.1. (C) *Cribroperidinium* sp., level AP-10.1. (D) *Ginginodinium* cf. *evittii*, level SJ-2.1. (E) *Tenua hystrix*, level SJ-2.1. (F) *Odontochitina rhakodes*, level SJ-3.1. (G) *Oligosphaeridium complex*, level SJ-2.1. (H) *Kiokansium unituberculatum*, level AP-13-1. (I) *Cycadopites* sp., level SJ-3.7. (J) *Afropollis jardinus*, level CA-5.2. (K) *Inaperturopollenites dubius*, level SJ-2.1. (L) *Uesuguipollenites callosus*, level AP-6. (M) *Callialasporites segmentatus*, level AP-6. (N) *Callialasporites trilobatus*, level AP-13.2. (O) *Gnetaceaepollenites oreadis*, level AP-16.2. (P) *Exesipollenites tumulus*, level SJ-2.1. (Q) Tetrad of *Classopollis major*, level SJ-2.3. (R) *Parvisaccites radiatus*, level AP-4.3. (S) *Vitreisporites pallidus*, level SJ-2.2. (T) *Podocarpidites* sp., level AP-25.6. Specimens A–C, H, L–O, R, T are from the Arroyo de la Pascueta section; specimens D–G, I, K, P–Q, S are from the San Just section; the specimen J was found in a sample from the Cortes de Arenoso section. Scale bar equals 20 μm except in S when it equals 10 μm.

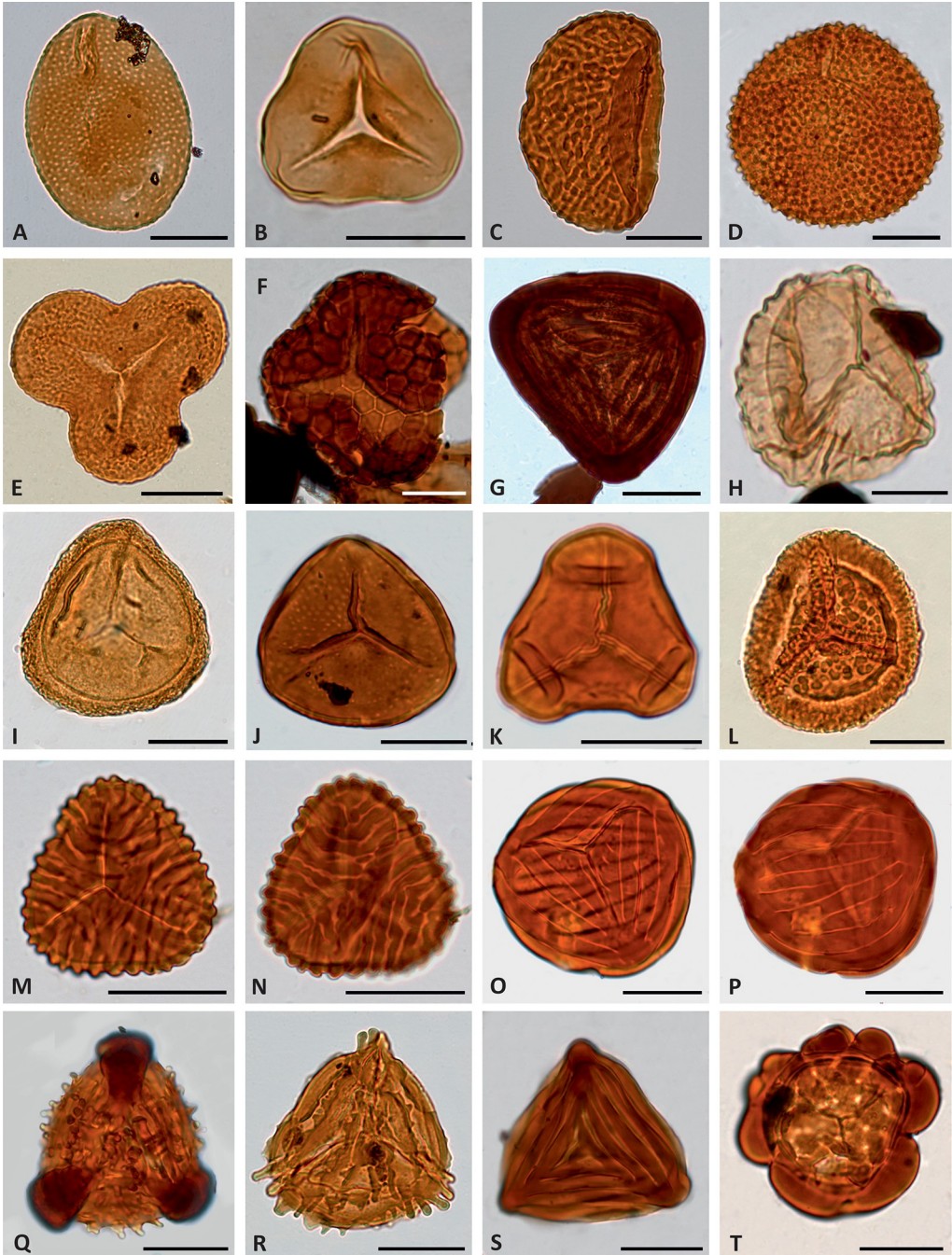

**Fig 8. Light photomicrographs of selected spores of Bryophyta and Pteridophyta.** (A) *Reticulosporis gallicus*, level CA-21.03. (B) *Cyathidites australis*, level SJ-2.1. (C) *Polypodiisporonites cenomanianus*, level CA-35.01. (D) *Verrucosisporites* sp., level AP-25.6. (E) *Concavissimisporites* cf. *crassatus*, level SJ-3.6. (F) *Gregussisporites orientalis*, level SJ-5.2. (G) *Contignisporites cooksoniae*, level AP-25.9. (H) *Staplinisporites caminus*, level AP-25.6. (I) *Densoisporites velatus*, level CA-2. (J) *Foveosporites* cf. *parviretus*, level AP-25.7. (K) *Cibotiumspora juriensis*, level CA-5.1. (L) *Taurocusporites segmentatus*, level SJ-3.1. (M–N) *Cicatricosisporites proxiradiatus*, level CA-21.04, (M) proximal side, (N) distal side. (O–P) *Cicatricosisporites potomacensis*, level AP-25.7, (O) proximal side, (P) distal side. (Q) *Nodosisporites segmentus*, sample AP-25.7. (R) *Appendicisporites* cf. *crenimurus*, level CA-10.01. (S) *Appendicisporites tricornitatus*, level AP-4.3. (T) *Patellasporites tavaredensis*, level SJ-5.2. Specimens A, C, I, K, M–N, R are from Cortes de Arenoso section; specimens B, E–F, L, T are from San Just section; specimens D, G, H, J, O–Q, S are from Arroyo de la Pascueta section. Scale bar equals 20 μm except in H when it equals 10 μm.

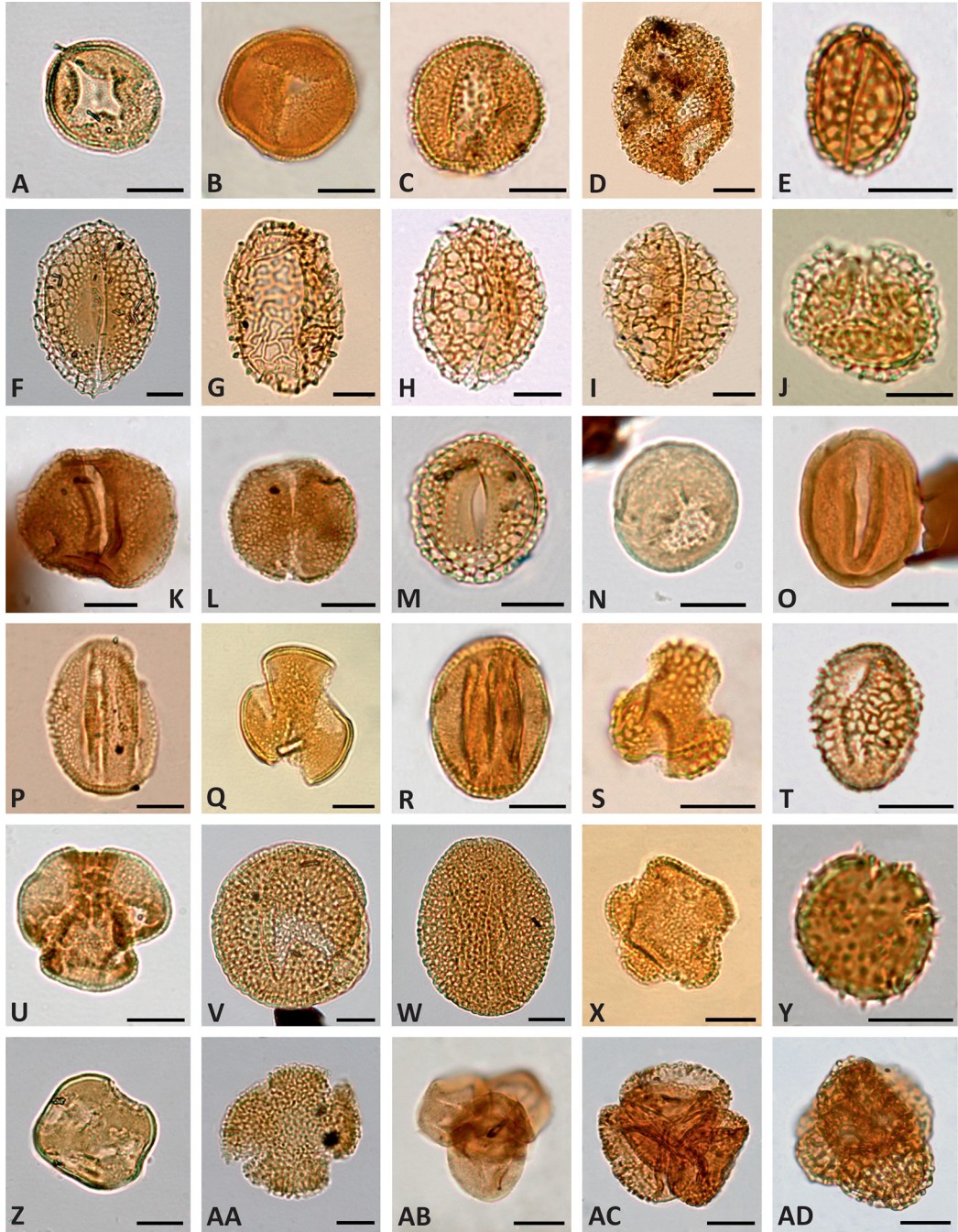

**Fig 9. Light photomicrographs of selected pollen grains of angiosperms.** (A) *Asteropollis asteroides*, level CA-2. (B) *Jusinghipollis* cf. *ticoensis*, level AP-10.2. (C) *Clavatipollenites tenellis*, level AP-32.3. (D) *Stellatopollis* cf. *barghoornii*, level SJ-5.2. (E) *Pennipollis reticulatus*, level CA-4. (F) *Liliacidites* cf. *clavatus*, level CA-14.01. (G) *Dichastopollenites* sp, level SJ-3.5. (H–I) *Dichastopollenites dunveganensis*, (H) level CA-7, (I) level SJ-3.8. (J) *Doyleipollenites robbinsiae*, level SJ-2.2. (K) *Retimonocolpites dividuus*, level AP-25.2. (L) *Dichastopollenites* cf. *reticulatus*, level AP-25.6. (M) *Liliacidites tectatus*, sample AP-25.7. (N) *Tucanopollis crisopolensis*, level CA-3.2. (O) *Transitoripollis* sp., level AP-25.6. (P) *Rousea georgensis*, level SJ-3.8. (Q) *Phimopollenites pannosus*, level SJ-3.7. (R) *Tricolpites nemejci*, level AP-10.2. (S–T) *Dryadopollis vestalis*, (S) specimen in polar view, level AP-10.2, (T) specimen in equatorial view, level CA-14.01. (U) *Phimopollenites pseudocheros*, level AP-34.2. (V–W) *Tricolpites* cf. *maximus*, (V) specimen in polar view, level CA-3.1, (W) specimen in equatorial view, level CA-3.1. (X) *Hammenia fredericksburgensis*, level SJ-3.4. (Y) *Echinipollis cenomanensis*, level CA-35.01. (Z) *Penetetrapites mollis*, level CA-1. (AA) *Hammenia fredericksburgensis*, level CA-35.01. (AB) *Artiopollis praecox*, level AP-25.2. (AC) *Senectotetradites* sp., level AP-10.2. (AD) *Senectotetradites grossus*, level CA-35.01. Specimens A, E–F, H, N, V–W, Y–AA, AD are from Cortes de Arenoso section; specimens B–C, K–M, O, R–S, U, AB–AC are from Arroyo de la Pascueta section; specimens D, G, I–J, P–Q, T, X are from San Just section. Scale bar equals 10 μm.

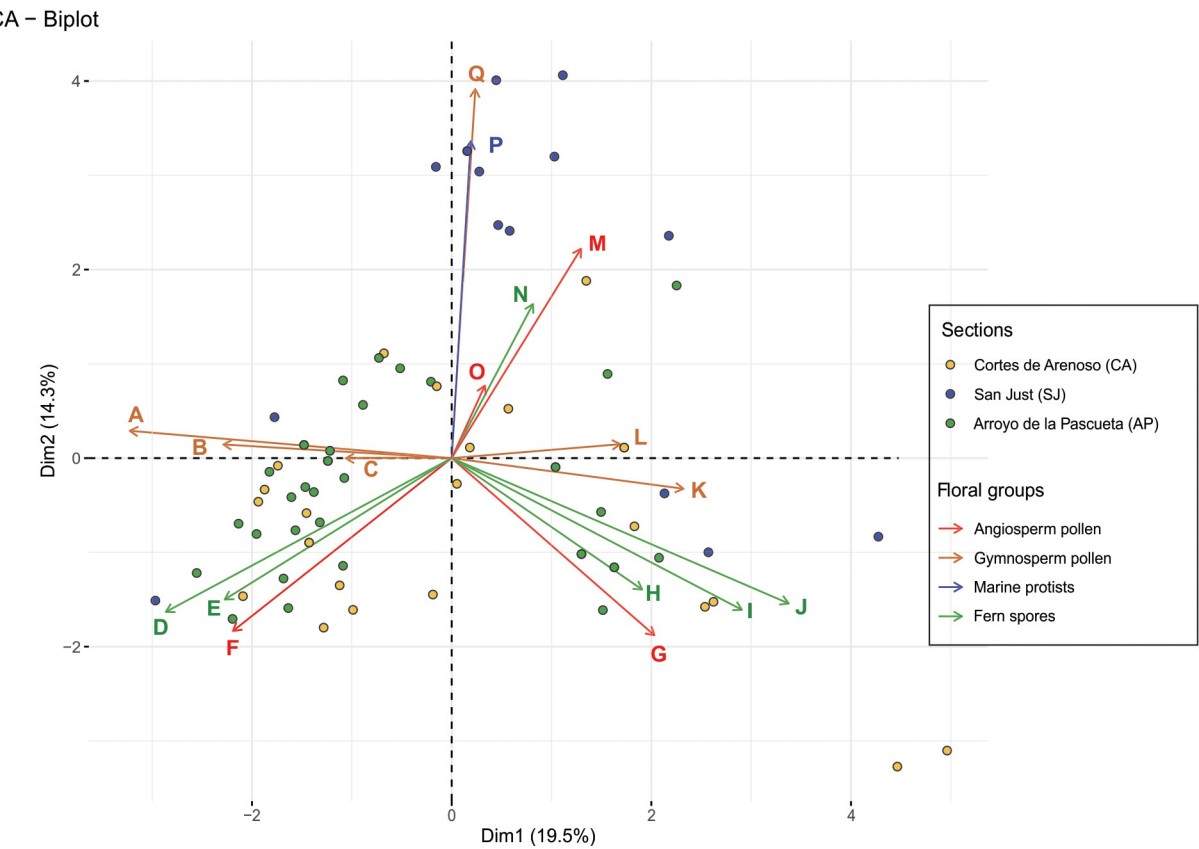

PCA − Biplot

**Fig 10. Principal Component Analysis (PCA) performed with the most relevant identified taxa and levels.** (A) *Classopollis* spp. (B) *Exesipollenites tumulus*. (C) Araucariaceae. (D) *Patellasporites tavaredensis*. (E) *Cicatricosisporites* spp. (F) *Crassipollis chaloneri*. (G) *Clavatipollenites* spp. (H) *Plicifera* spp. (I) *Cyathidites/Deltoidospora*. (J) *Gleicheniidites senonicus*. (K) *Monosulcites* spp. (L) *Eucommiidites* sp. (M) *Pennipollis* spp. (N) *Peromonolites allenensis*. (O) *Tucanopollis* spp. (P) Marine protists. (Q) *Inaperturopollenites dubius*.

and 7Q), constituting 80% of the total palynomorph sum at some levels. Other frequent gymnosperms include *Araucariacites australis*, *Exesipollenites tumulus* (Fig 7P) and *Monosulcites* spp. Bisaccate grains (Fig 7R–7T) usually represent a minor component of the assemblages. Fern spores represented the most diversified group of miospores in all the studied sections and occasionally reached dominance, being *Cyathidites australis* and *C. minor* (Fig 8B) the most frequently occurring species. Angiosperms, mainly represented by the genera *Clavatipollenites* (Fig 9C) and *Retimonocolpites* (Fig 9K), were recorded in significant numbers at specific levels from the CA and AP sections. A low number of tricolpate grains were identified in the three successions (Fig 9P–9W).

The PCA presents a variance of 33.8% in the first two dimensions (Fig 10). On the first axis (19.5% of variance), the taxa with negative coordinates were *Classopollis* spp., *Exesipollenites tumulus*, Araucariaceae, *Patellasporites tavaredensis*, *Cicatricosisporites* spp. and *Crassipollis chaloneri*. In contrast, *Inaperturopollenites dubius*, *Eucommiidites troedsonii*, *Peromonolites allenensis*, *Monosulcites* spp., *Cyathidites/Deltoidospora*, Gleicheniaceae, *Clavatipollenites* spp., *Pennipollis* spp., *Tucanopollis* spp. and the dinocysts presented positive coordinates in the PCA plot. The scattering of samples from AP and SJ along the first axis of the plot suggests the existence of an ecological gradient defining two vegetation types at the extremes. The second axis (14.3% of variance) discriminates spores in the negative extreme, from wind-transported gymnosperm pollen grains plus marine protists, in the positive one. While samples from AP and

CA are mainly plotted against the negative side of the axis, levels from SJ are mainly restricted to the positive quadrant.

Mega- and mesoremains were collected in the three studied sections. Generally, they exhibit good preservation and occur more abundantly in levels with marine influence (i.e., associated with palynological samples containing dinocysts) and without amber (e.g., SJ-2.1, CA-2, AP-4.2; Figs 3–5). In sharp contrast with the palynofloras, the megafloral assemblages are poorly diverse and dominated by conifers, being *Frenelopsis* particularly abundant. Angiosperms represent only a minor part of the mesofossil assemblages. The megaremain assemblage of AP has been interpreted as mostly allochthonous due to the range of disarticulation and fragmentation of specimens of *Frenelopsis turolensis* [106]. The preservation of most of the fossil plants as compressions including well-preserved cuticles and the occurrence of ramified and articulated branches of *Frenelopsis* (Fig 11D), some of them presenting cones attached to their axes, would indicate, however, a parautochthonous production *sensu* [107].

One of the most abundant taxon is *Frenelopsis*, which presents in the Maestrazgo Basin a vegetative morphology markedly xeromorphic with highly reduced leaves, thick cuticles and deeply sunken stomatas [37, 106]. Leaves of *Eretmophyllum*, *Mirovia* and *Dammarites*, as well as twigs of *Brachyphyllum* and *Widdringtonites*, have been commonly found.

**San Just section (SJ).**  The lower palynological interval of this section (SJ-2.1–SJ-3.4; Fig 12 clusters I–II) is characterised by marine palynomorphs and high percentages of *Inaperturopollenites dubius* (Fig 7K) and *Classopollis* spp. (Fig 7Q). Its upper part (SJ-3.5–SJ-5.3; Fig 12 clusters III–V), corresponding with the amber outcrop, is devoid of dinocysts and reveals increasing abundances of fern spores (*Cyathidites/Deltoidospora* and Gleicheniaceae) and angiosperms (mainly *Pennipollis*). A renewed increase of *I. dubius* and *Classopollis*, this time accompanied by numerous Anemiaceae and spores of uncertain botanical affinities (*Patellasporites tavaredensis*, Fig 8T), is characterising the uppermost interval of the succession (Fig 12, cluster V).

The megaflora collected in SJ (Fig 3) is the most diversified among the studied outcrops from the Maestrazgo Basin. *Frenelopsis justae* is numerically prevailing (Fig 11E). This species shows robust axes consisting of nodes and internodes formed by three unified leaves [37]. In addition, scarce remains of *Brachyphyllum* cf. *obesum* (Fig 11I), *Eretmophyllum* sp., *Dammarites* sp. and *Widdringtonites* sp. (Fig 11J) have been found. Several small fruits of angiosperms were found as mesofossils. They mainly include fruits of *Montsechia*-type (Fig 11F) and segmented fruitlets of cf. *Serialis* sp. (Fig 11H).

**Cortes de Arenoso section (CA).**  The lower palynological interval of CA (CA-1–CA-14.01; Fig 13, clusters I–IV) is characterised by fluctuating abundances of *Classopollis* and *Inaperturopollenites dubius*. The levels with the highest numbers of *Classopollis* usually include more dinocysts (*Cyclonephelium vannophorum* and *Kiokansium unituberculatum*) and other marine palynomorphs. In this interval, two levels with significantly high percentages of angiosperms (>40% of the total miospores) are noteworthy. The lower (CA-3.2) is characterised by the dominance of *Tucanopollis* spp. (Fig 9N), while the upper (CA-14.01; Fig 13, cluster IV) is characterised by a spike of "*Liliacidites*" *minutus*, and the near-complete absence of conifers. The overlying interval (CA-21.01–CA-35.03; Fig 13) includes high percentages of *Classopollis* (cluster V) and *Inaperturopollenites* (cluster VI), accompanied by moderate amounts of bisaccate grains and lower proportions of spores. The amber-bearing stratum (La Hoya amber outcrop) is restricted to the uppermost part of the succession (cluster VI).

Over 90% of the plant megaremains of this section are attributed to *Frenelopsis* cf. *turolensis* (Fig 11D). Particularly, this assemblage occurs in its lower part (levels CA-2, CA-3, CA-4, CA-10; Fig 5). Leaves of *Mirovia gothanii* and *Nehvizdya* (= *Eretmophyllum*) *penalveri* and elongated seeds of the *Spermatites*-type constitute the remains of the assemblage.

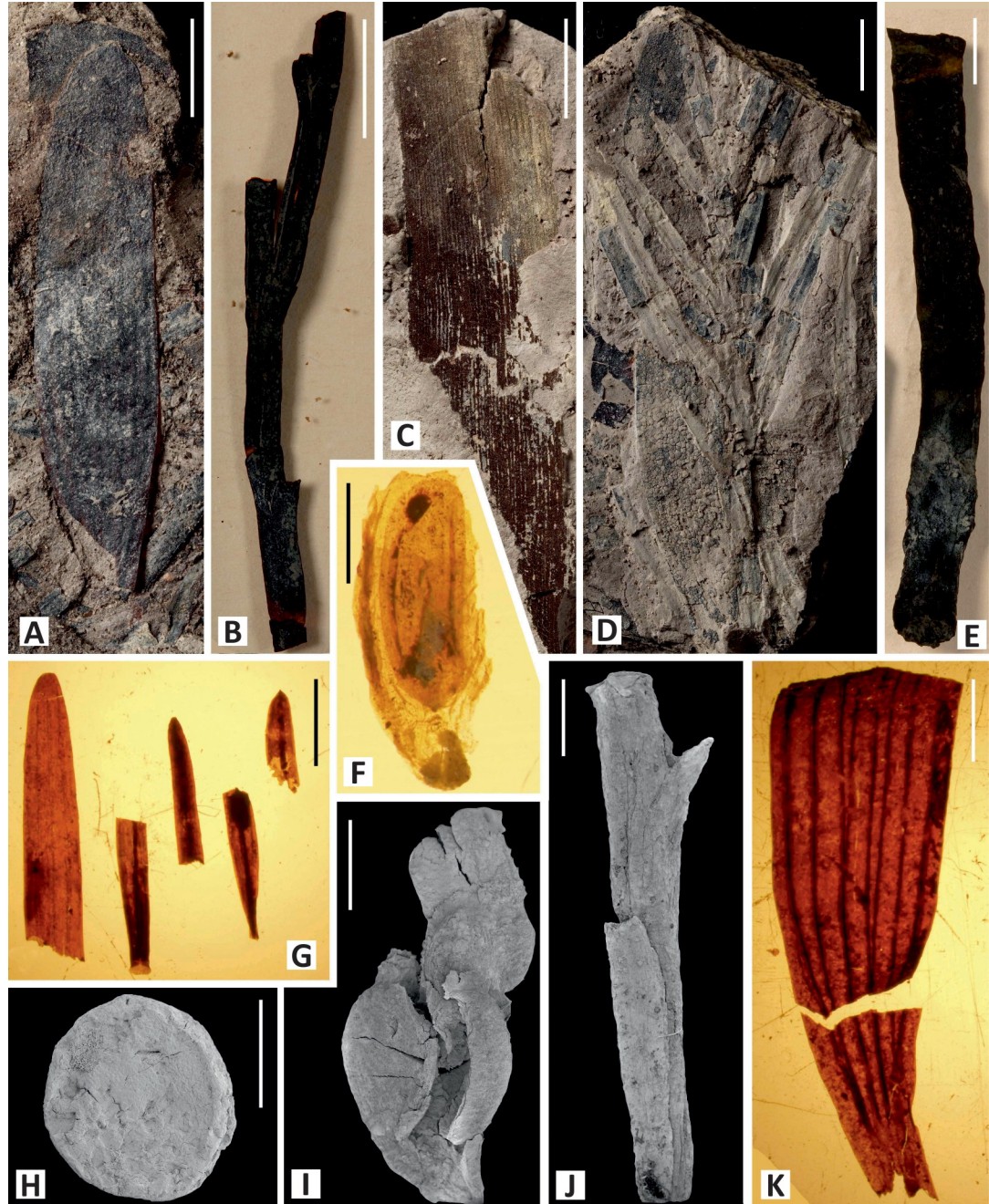

**Fig 11. Selected plant macro and mesofossils from the El Maestrazgo Basin.** (A) Leaf of *Nehvizdya* (= *Eretmophyllum*) *penalveri*, No. K 2788. (B) Shoot of *Frenelopsis turolensis*, No. K 2786. (C) Fragment of leaf of *Dammarites* cf. *albens*, No. K 2789. (D) Ramified shoot of *Frenelopsis* cf. *turolensis*, No. K 2797. (E) Shoot of *Frenelopsis justae*, No. K 2806. (F) Fruit fragment of *Montsechia*-type with orthotropous seed (*Spermatites*), No. K 2795. (G) Fragments of leaves of *Mirovia gothanii*, Nos. K 2807–2811. (H) Angiosperm fruit of cf. *Serialis* sp., SEM image, No. K 2803. (I) Shoot fragment of *Brachyphyllum* cf. *obesum*, SEM image, No. K 2804. (J) Shoot fragment of *Widdringtonites* sp., SEM image, No. K 2805. (K) Basal part of a leaf of *Nehvizdya* (= *Eretmophyllum*) *penalveri*, No. K 2794. Specimens A–C, G are from Arroyo de la Pascueta section; specimens D, F, K are from Cortes de Arenoso section; specimens E, H–J are from San Just section. Scale bars: A–D = 10 mm; E, G, K = 5 mm; F, H–J = 1 mm.

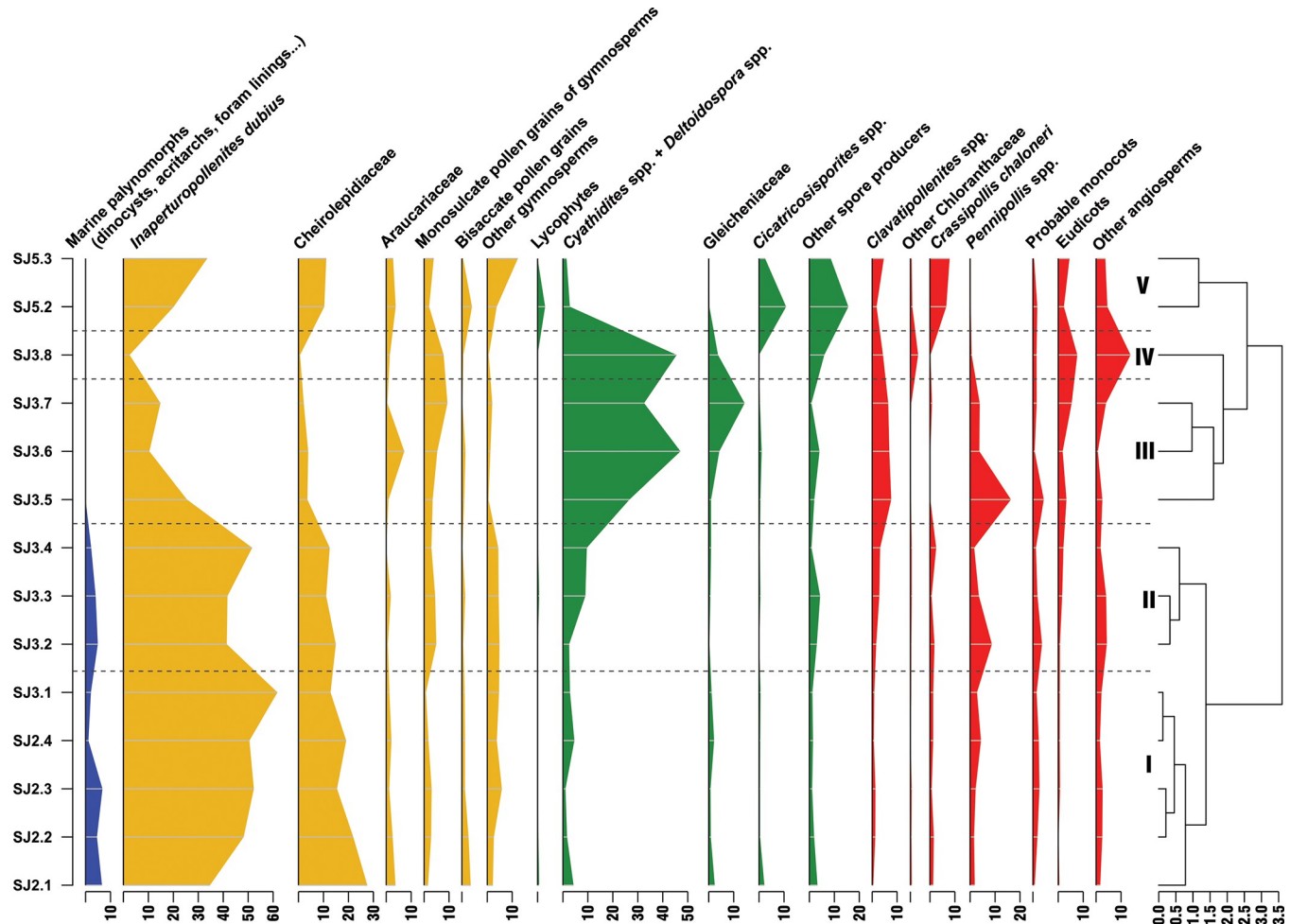

**Fig 12. Detailed palynological chart for San Just section (SJ).** Blue colour: marine palynomorphs; yellow colour: spores of ferns and allied; green colour: Gymnosperm pollen; red colour: Angiosperm pollen.

**Arroyo de la Pascueta outcrop (AP).** In the palynological succession, *Cyathidites/Deltoi-dospora* and *Classopollis* exhibit antagonistic percentage fluctuations. Cheirolepidiaceae characterised the lower part of the succession (Fig 14, clusters I and III). Conversely, although defining a pattern slightly less pronounced, *Cyathidites/Deltoidospora* are more abundant in the upper part of the section (Fig 14, cluster V and cluster VI), and occur in assemblage devoid of marine palynomorphs. The two mentioned intervals are separated by a level with abundant angiosperms and common Anemiaceae (cluster IV). The amber-bearing stratum, typified by *Cyathidites/Deltoidospora*, is restricted to the uppermost part of the succession (cluster VI). Although not found in abundance, angiosperms are diverse in the amber-bearing bed and include various species associated with Chloranthaceae/Ceratophyllaceae, Laurales and eudi-cots (i.e., Fig 9U).

Mega and mesofossil plants from AP show a balanced representation of remains with thick cuticles. The megaflora includes narrowly elongated leaves of *Mirovia gothanii* (Fig 11C), broadly obtuse entire-margined leaves of the ginkgophyte *Nehvizdya* (= *Eretmophyllum*) *penalveri* (Fig 11A), shoots of the Cheirolepidiaceae *Frenelopsis turolensis* (Fig 11B) and elon-gated leaves of *Dammarites* cf. *albens* (Fig 11C). While elongate seeds of the *Spermatites* type

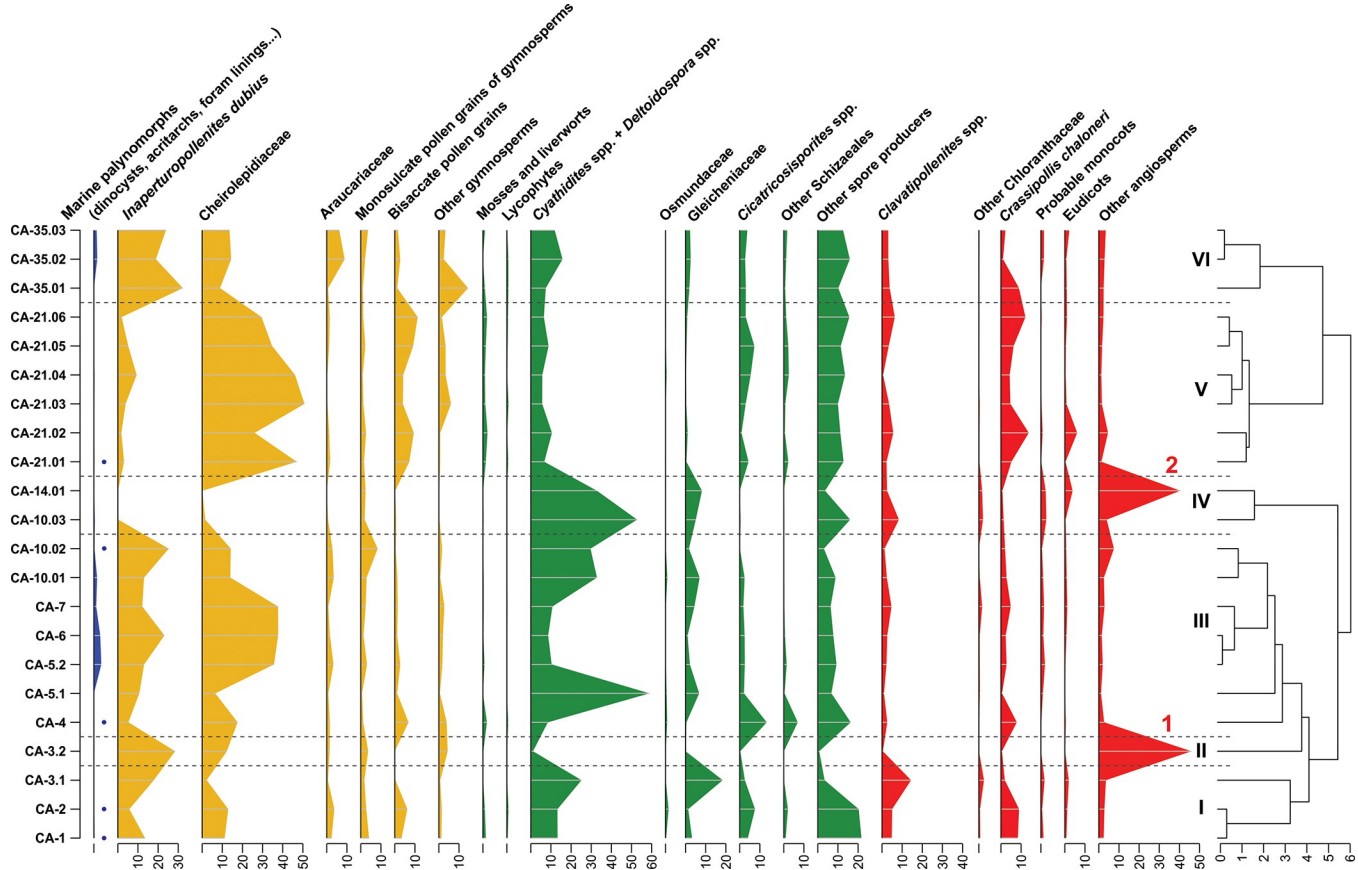

**Fig 13. Detailed palynological chart for Cortes de Arenoso section (CA).** Blue colour: marine palynomorphs; yellow colour: spores of ferns and allied; green colour: Gymnosperm pollen; red colour: Angiosperm pollen. 1. spike due to the abundance of *Tucanopollis* spp. 2. spike due to the abundance of "*Liliacidites*" *minutus*.

are common, the anecdotic presence of megaspores (two specimens) is noteworthy as it may reflect occasional floodings or high-water table allowing the development of wetlands.

## Insects and amber

The Cretaceous amber from the Maestrazgo Basin has provided an extensive arthropod record as bioinclusions from the San Just, Arroyo de la Pascueta and La Hoya outcrops (Table 2). San Just represents one of the most outstanding Cretaceous amber-bearing outcrops worldwide with diverse and abundant bioinclusions. Its entomological content has been the subject of extensive research over the last decades (i.e. [108, 109] and references therein).

The arthropod assemblages from San Just include arachnids (Pseudoscorpions, Acari and Araneae; Fig 4C), non-insect hexapods (Collembola) and up to eleven orders of insects (Archaeognatha, Orthoptera, Blattodea, Mantodea, Psocodea, Thysanoptera, Hemiptera, Neuroptera, Coleoptera [Fig 4D], Hymenoptera [Fig 4E and 4F] and Diptera), together with a high number of arthropod ichnofossils such as spiderwebs and coprolites. The site is also the type locality of 22 arthropod taxa to date, most of them insects (Table 2). Hymenoptera (mainly platygastroids [Fig 4E]) and Diptera are the most abundant insects. Apart from arthropods, the amber from SJ also includes dinosaur feathers (Fig 4F), although the outcrop lacks faunal megaremains. Conversely, the bioinclusions from the Arroyo de la Pascueta and La Hoya

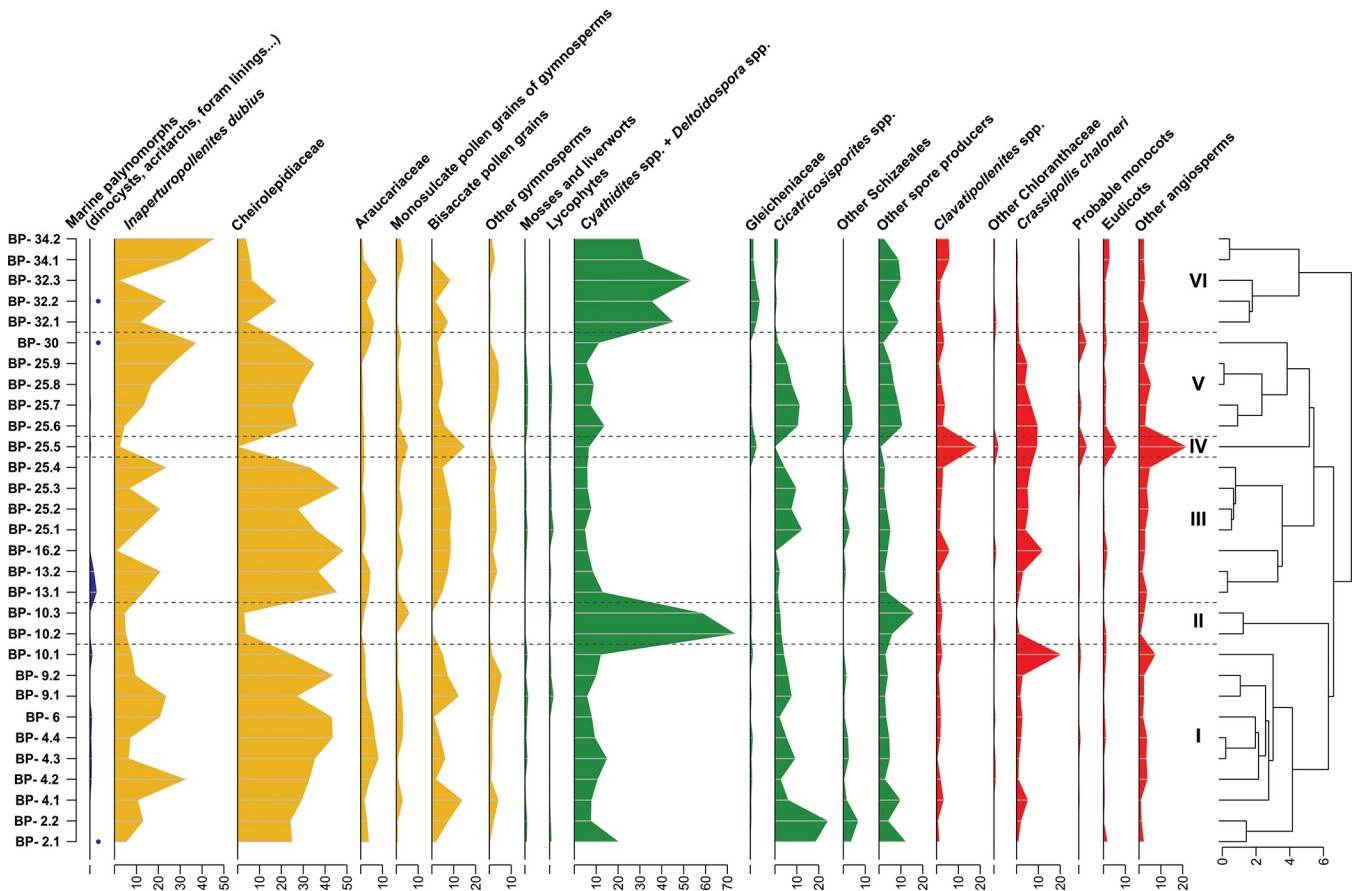

**Fig 14. Detailed palynological chart for Arroyo de la Pascueta section (AP).** Blue colour: marine palynomorphs; yellow colour: spores of ferns and allied; green colour: Gymnosperm pollen; red colour: Angiosperm pollen.

outcrops are poorly known and only few amber pieces have been described so far. Additionally, new material collected from these two outcrops is currently being prepared and new findings are expected. Hymenoptera, including one new species [110], Hemiptera and Diptera have been reported from Arroyo de la Pascueta while Diptera and Blattodea have been recorded in La Hoya. Collectivelly, the amber from the three outcrops has delivered 40 arthropod taxa, presenting higher or similar level of diversity than other mid-Cretaceous amber-bearing sites of Iberia such as Peñacerrada and El Soplao (northern Spain).

## Discussion

### Palynomorph-based biostratigraphy

In the AP section, the presence of *Tricolporoidites* (Fig 6), *Artiopollis praecox* (Fig 9AB), *Dryadopollis vestalis* (Fig 9S), *Gnetaceaepollenites oreadis* (Fig 7O), *Liliacidites tectatus* (Fig 9M), *Rousea miculipollis*, *Senectotetradites* spp. (Fig 9AC) and *Tricolpites nemejci* (Fig 9R) indicates a late Albian age (Fig 15) since none of these taxa have been identified in older strata [78, 93, 104, 111–113]. This age is significantly younger than the one inferred by [29] based on a long-distance correlation with palaeontological data from the neighbouring Morella Sub-basin.

The age of the SJ section can be independently dated as late Albian or younger based on the record of *Odontochitina rhakodes* (Fig 7F). This age is further supported by the presence of

**Fig 15. Composite range chart showing selected species of biostratigraphic interest identified in the Lower Cretaceous sections from the Maestrazgo Basin.** L = lower; M = middle; U = upper.

*Tricolpites nemejci* and *Senectotetradites* sp. [93, 104, 111, 114–116]. This age inference updates and refines the previously proposed middle to early late Albian age, which was based on a more limited palynological data set [32]. The presence of *Cicatricosisporites apicanalis*, *Contignisporites cooksoniae* (Fig 8G), *Gregussisporites orientalis* (Fig 8F) and *Nevesisporites dailyi* (Fig 15) in both SJ and AP, also allows defining an intra late Albian upper limit for the age interval as none of the taxa have been recorded in strata of different age in Europe [31, 113, 117].

In addition to the biostratigraphic markers already present in SJ and AP, the lower part of the CA section also includes *Reticulosporis gallicus* (Fig 8A), a taxon only recorded in France in strata independently dated as late Albian [90, 118], and *Microfoveolatosporis pseudoreticulatus* and *Cupuliferoidaepollenites parvulus* (Fig 15), both taxa presenting the oldest occurrence in strata of that age in North America and Western Europe [105, 112, 119]. A slightly younger early Cenomanian could be considered for the uppermost part of the CA section that corresponds to the La Hoya amber outcrop (CA-35.1–CA-35.3; Fig 5) as *Echinipollis cenomanensis* (Fig 9Y), *Polypodiisporonites cenomanianus* (Fig 8C) and *Senectotetradites grossus* (Fig 9AB) recorded therein have not been reported from older strata [111, 120–122]. The CA record

could equally represent a range extension down into the late Albian for these taxa, which remain not widely distributed (i.e., rare species).

The updated new age for the sections has important implications for the local stratigraphy as it allows us to confidently attribute them to the Utrillas Group [61], while previous studies considered them as part of the Escucha Formation [28, 29, 31, 32, 43]. The new biostrati-graphic inferences are also supported by previous results obtained for the Utrillas Group in the Maestrazgo Basin based on indirect correlations [47, 50], palynological [123, 124] and faunal data (ostracods) [61].

From a palaeoenvironmental perspective, the biostratigraphic markers recorded in the three studied sections suggest that resin production/deposition in the Maestrazgo Basin was either coeval or close in time. In that regard, the amber levels might represent a biostrati-graphic event of regional significance. The marine carbonates of the Mosqueruela Formation [50], immediately overlying the amber-bearing strata at CA and AP, represent the regional expression of the flooding event described elsewhere in uppermost Albian strata (Mortoni-ceras rostratum/M. perinflatum ammonite zones) [125]. In eastern Iberia, this transgressive pulse led to the development of widespread marine platforms across the Mesozoic intraplate basins [48, 49].

## Environments and vegetation

The record of highly variable fossil remains and amber with biological inclusions suggest the presence of an ecosystem structurally framed by multi-tiered vegetation [14, 28, 34–36] sus-taining complex arthropod communities (Table 2). Terrestrial ecosystems of such complexity might seem surprising for deposits associated with a desert system. However, there are no rea-sons to believe that arid and semi-arid ecosystems from the Cretaceous were less complex than their modern counterparts, such as the Namib and Kalahari deserts [126, 127]. The presence of palynological assemblages of varying composition in marine-influenced intervals is other-wise expected, given the taphonomic, palynological and palaeobotanical constraints [14, 128–131], and indicates the existence of mosaic vegetation types in the area with conifer-forests along the Tethyan coasts (Fig 16). The vegetation of this region was strongly conditioned by both the influence of the sea and the desert and other factors such as palaeowildfires, as the abundant charcoal occurrence in the Spanish Cretaceous amber-bearing deposits indicates [40, 132].

## A picture of the regional vegetation: Cheirolepidiaceae-Cupressaceae communities

The significant representation of *Classopollis* and *Inaperturopollenites dubius* in the studied assemblages reflects the importance of the families Cheirolepidiaceae and Cupressaceae in the vegetation. The increased numbers of these two taxa observed in assemblages with marine palynomorphs (Figs 12–14) and the correlation between *I. dubius* and dinocysts in the PCA (Fig 10), representing the most distal settings indicate that their pollen-producers were wide-spread regionally, probably forming a significant part of the hinterland vegetation.

Cheirolepidiaceae was one of the most widespread conifer families among the Mesozoic vegetation. They colonised a wide range of environments ranging from low to high latitudes in both hemispheres [8, 133–135] and commonly assumed a dominant position in the vegetation from the Early Jurassic to the mid-Cretaceous [134, 136], especially in some low latitude set-tings associated with evaporites [8, 137]. While historically linked to arid environments and/or salty ones [79, 138–140], recent studies suggest broader ecological preferences for, at least, some representatives of the family [134, 141, 142].

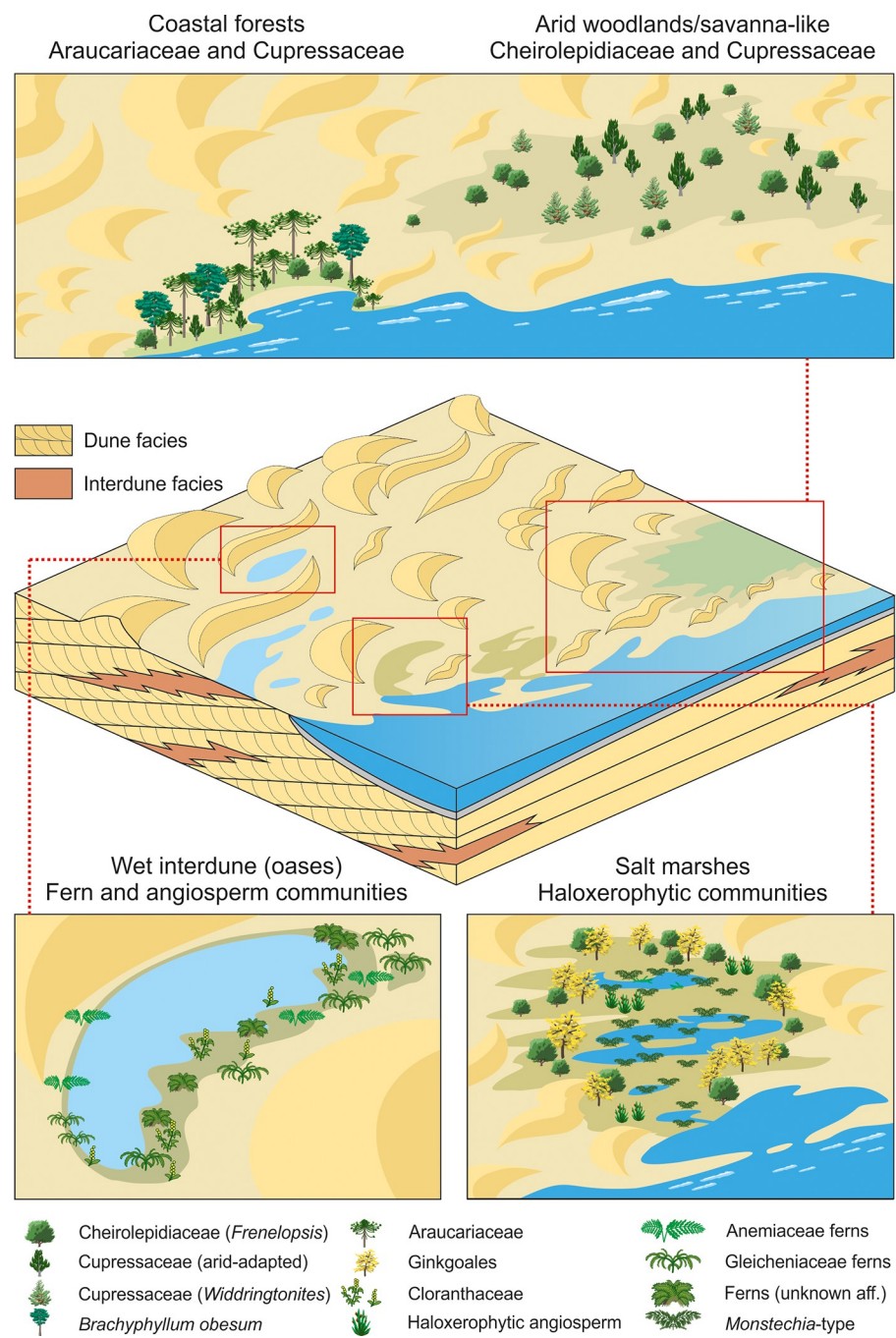

**Fig 16. Schematic model of the late Albian fore-erg setting, including the types of vegetation and their distribution in the different sub-environments.**

*Inaperturopollenites dubius*-producers are thought to belong to Cupressaceae, a conifer family with species widely distributed in present-day ecosystems of the Northern Hemisphere [143]. While some modern forms, such as the bald cypress and the Chinese swamp cypress, mostly belonging to the *Glyptostrobus–Taxodium–Cryptomeria* clade (Taxodioideae), exhibit a marked ecological affinity toward wet environments [144], a significant number of Cupressaceae species prefer well-drained habitats [143]. Distinguishing moisture-loving forms from

plants preferring well-drained substrates based solely on palynological evidence is complex due to the similarity of the pollen produced by both groups of plants. By analogy with modern pollen of *Taxodium*, fossil grains presenting a papilla such as *Taxodiaceaepollenites* are assumed to have been produced by trees preferring saturated soils and wetlands subjected to periodic floodings. Taxodioideae pollen is frequent in Cenozoic deposits and has also been found in high numbers in Cretaceous deposits from high latitude locations [135, 145]. These pollen grains have also been recorded in mid-latitude assemblages associated with spores produced by water ferns [90]. However, they are scarcely represented in the successions from the Maestrazgo Basin (S1 Appendix). *Perinopollenites halonatus*, pollen with two distinct exine layers related to Taxodioideae [77], neither reaches a significant representation in the studied sections (S1 Appendix). Therefore, most Cupressaceae pollen recovered is assumed to have been produced by trees or shrubs that preferred well-drained substrates.

The presence of *Widdringtonites* sp. in SJ (Fig 11J) would support the presence of Cupressaceae linked to well-drained substrates as the specimens assigned to this taxon present well-developed Florin rings in their stomata, reflecting xerophytic conditions. Interestingly, the genus *Widdringtonites* has been related to the extant *Widdringtonia*, which presents species adapted to very hot, dry summers and recurring wildfires in South Africa [146].

If an arid/semi-arid environment can be inferred from the large number of Cupressaceae pollen and *Classopollis* as well as regional sedimentology, the presence of localised thickets of Taxodioideae colonising the wettest parts of the landscape (i.e., wet interdunes, see below), while unlikely, cannot be discarded. The palynofloral data presented here and indicating the prevalence of Cupressaceae in local vegetation confirm the results obtained from the fossil wood dataset for the Albian (T2 time bin in [5]).

Cheirolepidiaceae and Cupressaceae are often the co-dominant families in the studied palynofloras. Fluctuating frequencies of these taxa could, however, reflect drastic changes in vegetation which are difficult to ascertain. While Cheirolepidiaceae usually dominate in AP and CA (Figs 13 and 14), Cupressaceae remains are more abundant in SJ palynofloras and the amber-bearing levels of AP and CA (Figs 12–14). These two taxa have previously recorded similar dominance of the palynofloras in numerous mid-latitude Jurassic and Cretaceous deposits [25, 90, 104, 136, 142]. Present-day growth habits of Cupressaceae, as well as whole plant reconstructions of fossil representatives of Cheirolepidiaceae, suggest that these plants were dominantly, if not exclusively, arborescent during the Cretaceous. The corresponding shrubs and trees might have represented the sub-dominant and dominant components of open vegetation (savanna–woodlands; Fig 16) similar to those observed in modern arid to semi-arid climates [143]. This conifer-dominated plant community was very different from the Jurassic and Cretaceous fern savanna biome characterised by the families Anemiaceae and Matoniaceae [75, 147–151] which might have also grown on dunes.

The diversity of angiosperm pollen types and their varying abundance pattern (S2 Appendix) suggest that producers had distinct ecophysiologies allowing them to colonise not only moister habitats and, probably, water-logged, if not aquatic, environments [12, 35, 152–154] but also well-drained and/or dry edaphic settings. Originally associated with moist environments [155], angiosperms may have already radiated into more arid settings by the late Albian, as their presence in deposits associated with the desert system could suggest [14]. It is likely that the representation of flowering plants in pre-Cenomanian ecosystems is underestimated owing to morpho-functional and taphonomic biases (i.e., low production and dispersal of entomophilous pollen [156]). While the exact phenology of early angiosperms is poorly understood, the existence of short life cycles allowing flash flowering [152, 157] could explain the localised high abundance of this pollen.

### Diversified local vegetation.

- Coastal settings with Araucariaceae

Pollen of Araucariaceae is consistently recorded in the palynofloras of the three studied outcrops (S1 Appendix), albeit in low abundance (<5% of the total miospores). However, the presence and relative abundance of these conifers in the local vegetation may be underestimated since both modern [158, 159] and fossil Araucariaceae pollen [136] are believed to present low dispersal capabilities. Several researchers [158] noted that modern palynological assemblages collected from Kauri (*Agathis australis*) forests only include relatively low percentages (<5–10%) of Araucariaceae pollen.

Palaeobotanical and geochemical evidence suggests a clear Araucariacean affinity for the late Albian amber recovered in the Basque-Cantabrian Basin [90, 160]. However, a geochemical analysis of the amber from the San Just outcrop seems to support an araucariacean-cheirolepidiacean affinity [161]. The ubiquity of Araucariaceae in palynological assemblages and the record of presumed specialised insect pollen feeders (nemonychid beetles) in amber inclusions from the San Just outcrop [162] would represent additional circumstantial evidence linking Araucariaceae to the resin production.

The precise role of Araucariaceae in Cretaceous vegetation and the structure of the forest types they formed still need to be ascertained, given the characteristics of the habitats colonised by modern forms. Apart from isolated occurrences in savanna-like vegetation growing on mafic-ultramafic soils of New Caledonia (i.e., *Agathis ovata*; [163]), modern araucarians are often part of evergreen rainforests where they act as emergents (*Agathis microstachya* in Australia, *A. ovata* and *Araucaria schmidii* in New Caledonia, *Agathis australis* in New Zealand), or constitute small thickets and discrete stands in low altitude forests of New Zealand (*A. australis*) and Australia (*Araucaria heterophylla*) [164, 165]. Species distributed in low-fertility soils, ridgetops, and steep slopes also reflect probable competitive exclusion by angiosperms, i.e., a derived, relictual distribution (a realised niche much more restricted than a potential niche sensu [166]). Some Recent Araucariaceae, such as *Araucaria columnaris*, *A. luxurians* and *A. nemorosa* are coastal trees able to withstand the influence of salt wind [74, 167], and this ecological habit was probably already exhibited by representatives of the family during the Mesozoic. The presence of *Microphorites utrillensis* (Dolichopodidae: Microphorinae) in amber from the SJ outcrop [168] would support a sandy coastal habitat for *Araucariacites*-producers, as living microphorines are known to inhabit this type of habitats [169] (Figs 16 and 17).

It is interesting to mention the association of modern Araucariaceae with other conifers, mainly Podocarpaceae, but also Cupressaceae in the evergreen forests of New Zealand [165]. Associations among conifers, probably fortuitous within the context of modern vegetation dominated by angiosperms, should have been more frequent during most of the Mesozoic. The association of Araucariaceae with Cheirolepidiaceae and *Exesipollenites tumulus* in PCA (Fig 10) would support the existence of mixed-gymnosperm communities in the area. The spatial proximity of conifers from distinct families is also suggested by the megaremain assemblages where *Brachyphyllum* cf. *obesum* (Fig 11I), a species that indicates mesophytic to xerophytic conditions attributed to Araucariaceae, has been found together with the Cheirolepidiaceae *Frenelopsis justae* (Fig 11E).

Two types of plant formations involving Araucariaceae can be hypothesised: (i) thickets of limited lateral extent involving one or a few species of Araucariaceae as the dominant elements. This hypothesis, still not supported by *in situ* palaeobotanical evidence, could explain the low dispersion of pollen (i.e., underrepresentation in palynological assemblages), the

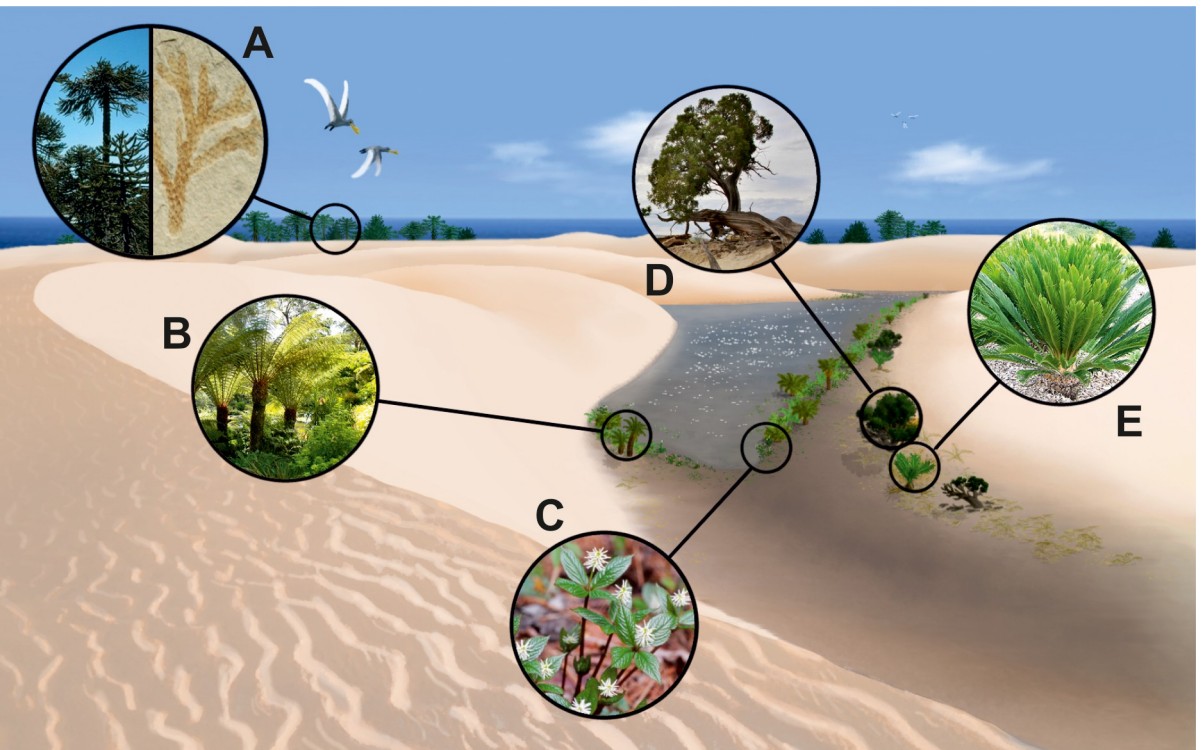

**Fig 17. Conceptual palaeoartistic reconstruction of the late Albian wet interdunes in the fore-erg from the Maestrazgo Basin (eastern Spain).** This environment developed near the Tethys, where (A) coastal forests with Araucariaceae grew. The riparian places of the wet interdunes (oasis) (B) were covered by fern and angiosperm vegetation. (C) Fresh water/brackish aquatic angiosperms inhabited wet interdunes. (D–E) Xerophytic plant community involving Cheirolepidiaceae, Cupressaceae (D) and other gymnosperms groups such as Bennettitales, Cycadales (*Monosulcites*/*Cycadopites*-producers) and Gnetales (*Equisetosporites*-producers) (E) that integrated arid woodlands.

resulting canopy acting as a barrier to pollen transport in a fashion analogous to modern stands of *Araucaria columnaris* in New Caledonia and *A. angustifolia* in Brazil [159] and a predicted poor species richness for the stand [170]; (ii) mixed conifer communities with Cupressaceae, Podocarpaceae or/and Cheirolepidiaceae conifers constituting single-tiered or two-tiered woodlands, the latter with emergent araucarians. It would be reasonable to assume the existence of a transition from woodlands to more open, savanna-like, vegetation structures constrained by local edaphic conditions and the presumed arid to semi-arid conditions at a regional scale.

The presence of Araucariaceae in hinterland vegetation could also be regarded as plausible, but it is likely not to have produced the recovered palynological signal given the low transport capacity of the araucarian pollen. Conversely, the ubiquity of Cheirolepidiaceae in pollen assemblages suggests a more efficient dispersal and that their contribution to the mixed conifer association described above might not have been that significant.

• Salt marshes and swamps

Mega and mesofloral assemblages from the Maestrazgo Basin are characterised by abundant *Frenelopsis*. While the genus has been associated with brackish coastal marshes when found in the mid-Cretaceous of Europe [106, 171, 172], it seems likely that the constituting species exhibited more diverse palaeoenvironmental affinities. In the three outcrops, the genus is represented by *F. turolensis* (Fig 11B), and/or *F. justae* (Fig 11E). While *F. turolensis* has

been reconstructed as a small ligneous plant or shrub growing in coastal salt-influenced environments [106, 173], *F. justae* [37] suggests more mesic settings [174, 175].

The association of *F. turolensis* with *Nehvizdya* (= *Eretmophyllum*) *penalveri* [29] (Fig 11A and 11K), a ginkgoalean previously interpreted as halophytic [83, 176], supports colonisation of salt-influenced settings (Fig 16). This type of ginkgoalean constituted coastal vegetation associated with Cheirolepidiaceae [177]. These communities could thrive in supratidal coastal areas [174, 178]. A similar inference has also been proposed by [28, 29] based on specimens collected in the AP succession. If not taphonomically induced, the low diversity of *Frenelopsis-Eretmophyllum* could be interpreted as supporting evidence of stressful ecological conditions [179].

Similarly, the complex nature of the fruit of *Montsechia*-type with *Spermatites* seeds inside recorded in SJ and CA (Fig 11F) could reflect near-shore deposits as these seeds (*Spermatites*) have been recorded in physical association with *Pseudoasterophyllites cretaceous*, a Cenomanian angiosperm described in tidally-influenced deposits from Bohemia [98, 180, 181]. While still debated, *Montsechia* has been considered by some authors as a halophytic plant growing in periodically desiccated coastal marshes [182]. Irrespective of the specific ecological requirements of *Montsechia*, the existence of halophytic/xerophytic vegetation formation, including angiosperms is supported by the presence of *Tucanopollis*. While this pollen has been recorded in European mid-Cretaceous deposits [183–185], it constitutes an essential element of assemblages characterising arid settings of central [186, 187] and NE Gondwana [16] and has been observed in stamens of *P. cretaceous* [98]. The high proportion of *Tucanopollis* in the lower part of the CA succession (representing the bulk of the 'other angiosperms' in level CA-3.2; Fig 13, cluster II; S1 Appendix) is assumed to reflect the proximity of a halophytic/xerophytic plant community also involving Cupressaceae (*Inaperturopollenites dubius*-producers), Ertmanithecales and Cheirolepidiaceae (*F. turolensis*) and probably *Monosulcites*-producers of uncertain botanical affinities (Fig 17; S1 Appendix). This plant association would characterise coastal, salt-influenced, episodically (but not periodically) flooded areas. The existence of mangroves (i.e., vegetation colonising habitats under *periodic* marine water influence) is possible, although currently not supported by anatomical evidence.

• Wet interdunes and wetlands

The presence of freshwater bodies in the study area during the late Albian to early Cenomanian has been inferred from sedimentological and palaeobotanical evidence [188, 189]. The existence of habitats with fluctuating humidity or with high moisture is also indicated in San Just by amber inclusions, including the hemipteran *Iberofoveopsis miguelesi* [44], as well as the presence of edaphic fauna linked to the litter degradation and soils [190–192] (Table 2).

The high proportions of fern spores in palynological assemblages devoid of marine palynomorphs and characterised by low numbers of gymnosperms (see Fig 12 clusters III and IV; Fig 13 cluster IV; Fig 14 clusters II and VI) also indicate that freshwater was available, at least during part of the year, as spore producers need free water to complete their biological cycle. A series of fern spikes, mainly consisting of abundant spores with smooth exine attributed to *Cyathidites* and *Deltoidospora*, are observed in the three sections.

The palynological signal corresponding to these spikes is challeging to interpret. Similar spikes have been recorded after the end-Permian, Triassic, and Cretaceous events [193–195] and interpreted as fern-dominated pioneer vegetation characterising the early stage of recovery after the disruption of some terrestrial ecosystems. If the existence of a global adverse palaeoenvironmental event during the late Albian can be discarded on available evidence, the fern spikes observed in the three sections could still be interpreted as riparian early successional developments [196] in the present context. Modern arid ecosystems are fragile and

prone to significant vegetation shifts when disturbances such as draughts occur [197, 198], and similar disruptions of vegetation could well have happened in similar past environments. A recovery vegetation reduced to pioneer ferns, and conifers and other resilient floral elements remains a plausible ecological scenario. It is interesting to highlight that palynofloras characterised by a high proportion of *Cyathidites* and *Deltoidospora* (94% of the total miospore content) have already been recorded in other deposits associated with the Iberian Aptian–Albian desert system in SW Spain (Oliete Sub-basin) [2, 199]. If the palaeoenvironmental interpretation presented here is valid, the recurring fern spikes suggest that episodes of accentuated draught were a common feature during the mid-Cretaceous at mid-latitude locations and that the vegetation responded to palaeoenvironmental disturbance in a similar way. The high content in spores of palynofloras reflecting arid and disturbed vegetation may appear counterintuitive, but a significant number of modern spore-producers can withstand extended periods of drought, present a high tolerance to environmental stress or simply thrive in edaphically low-nutrient habitats [200–202]. This adaptability has also been inferred for several Palaeozoic and Mesozoic spore-producers (e.g. Pennsylvanian sphenophytes, Jurassic Gleicheniaceae, Late Cretaceous Matoniaceae [73, 151]), which are often presented as stress-tolerators. Periodic ecological disturbances are also thought to have been favoured by early angiosperms [155]. Besides *Tucanopollis*-producers, angiosperms colonising the Iberian Desert System included representatives of the ANITA grade, Chloranthales of diverse affinities (*Clavatipollenites*, *Transitoripollis*, etc.), probable monocots (*Monocolpopollenites*, *Retimonocolpites*) and eudicots (Table 1). These angiosperms were probably well integrated into the landscape, forming part of multiple vegetation types, including terrestrial wetlands (wet interdunes, oases *sensu* [189]) and freshwater aquatic ecosystems as helophytes and/or floating elements [4, 34, 36] (Fig 17).

The occurrence of a high abundance of "*Liliacidites*" *minutus* in CA (Fig 13 clusters IV) could also reflect advanced stages of ecological succession, involving colonisation of settings dominated by ferns. This pollen type, currently related to Laurales [203], could have been produced by angiosperms of broad ecological tolerances similar to the one exhibited by *Eucalyptolaurus deprei*, a representative of the family described from the Cenomanian of France [176, 204].

The high number of spores attributed to Anemiaceae (i.e., *Cicatricosisporites*) in the studied assemblages indicates that ferns were probably still the dominant floral elements in wetlands and maybe in some arid places. Early Cretaceous representatives of the family have been previously associated with swamps, lacustrine, and savanna environments [75, 150, 205], and a similar environment can be assumed for the Iberian Anemiaceae, along with another spore (*Patellasporites tavaredensis*) and pollen (*Crassipollis chaloneri*) producers (Fig 10). The inferred vegetation should have been of low stature and have surrounded ephemerous or semi-permanent water bodies developed in interdunes, maybe as part of the understory of open woodlands [75, 188, 206].

- Hinterland vegetation

The presence of bisaccate grains in assemblages otherwise characterised by marine palynomorphs suggests that a fourth type of vegetation involving conifers could have constituted a more distant vegetation type. Bisaccate pollen grains present a high dispersal capability, and their higher representation in distal, marine settings is often interpreted to reflect a taphonomic (i.e., biostratinomic) bias [207, 208], known as Neves Effect in Palaeozoic floras [209]. In the present context, the higher abundance of bisaccate grains in marine assemblages could represent hinterland vegetation involving Pinaceae and/or Podocarpaceae that would have colonised topographically higher settings of the Iberian Massif [14].

## Comparison with other localities

The palynological assemblages from the Maestrazgo Basin documented here are similar to coeval successions from the Las Parras and Oliete sub-basins [31, 33–35]. *Afropollis* (Fig 7J), ephedroid polyplicate pollen (Fig 7O) and several angiosperm taxa such as *Dichastopollenites* (Fig 9G–9I and 9L) and *Cretacaeiporites* relate the Spanish assemblages to the palynofloras from North Gondwana (see i.e. [210–214]) and support Iberia as a transitional zone between the floral provinces of NE Gondwana and Laurasia [104]. The Maestrazgo palynological assemblages can also be related to the ones described from the Basque-Cantabrian Basin [31, 87, 104, 132] as both basins share more than 90% of the identified miospore families and a significant number of their respective species. The faunal record, particularly the arthropod assemblages characterising the two Iberian areas, reveals a series of common species [162]. Arthropods recorded in the Maestrazgo with a more widespread distribution include *Cretevania* (Hymenoptera: Evaniidae) documented in various outcrops across Europe and Asia [215], *Serphites* (Hymenoptera: Serphitidae) recorded in Europe, Asia, and North America [216] and *Microphorites* (Diptera: Dolichopodidae) found in ambers from France, Lebanon and Myanmar [168].

The coeval miospore assemblages from the Lusitanian Basin exhibit higher angiosperm abundance than the ones described herein [6, 105, 113, 217]. While this difference could likely reflect vegetation with more angiosperms, the higher representation of dinocysts of the Portuguese successions suggests the influence of the depositional environments as sedimentary successions on a passive margin can collect miospores from a larger catchment area.

The megafloral record relates the flora from the Maestrazgo to Laurasian assemblages. Taxonomically similar, low-diversity plant assemblages have been documented from mid-Cretaceous strata of southwestern and northern France [218–220] and the Czech Republic [83, 172]. These shared characteristics are interesting as they suggest that a few species of angiosperms were thriving in salt-influenced environments and that geographic and climatic barriers did not prevent a widespread distribution across Laurasia. Pathways allowing the dispersal of taxa with such specialised ecological requirements could include coastal fringes surrounding the Tethys. The good representation of *Tucanopollis* in both NE Gondwana and southern Laurasia suggests that a significant number of halophytic angiosperms used this 'coastal pathway' to colonise new environments. However, more direct, NS trending migration routes following northern and southern Atlantic rift valleys could also be considered to link central Gondwana floras of Gabon and Brazil to Laurasian floras.

Despite the numerous similarities existing between floras from Gondwana, mid and northern Laurasia, it is probable that the latitudinal migration of flora remained hindered by differences in climate. Floras from locations at equivalent palaeolatitudes, should always present more compositional and structural similarities than floras from distinct climatic belts. Late Albian–early Cenomanian floras from NW Europe and N America present higher percentages in Gleicheniaceae and *Cyathidites* but lower diversity in angiosperms than coetaneous floras from southern Europe.

Latitudinal effects influenced the composition of the mid-Cretaceous palynological assemblages from Western Europe and the Atlantic Coastal Plain of USA, since they belonged to the Northern Mid-latitude Warm humid belt (NMW belt) [17, 221]. Although the diversity of taxa was high in the palynological assemblages attributed to the NMW belt, the abundance of miospores is different from that of the Maestrazgo Basin (i.e., [90, 112, 222–226]), which was under the influence of the Northern Hot Arid (NHA) belt [17, 188] during the late Albian. More concretely, under the influence of the NMW belt, fern spores were generally abundant, showing conspicuous amounts of Gleicheniaceae and *Cyathidites*, the amounts of *Classopollis* indicate

that it was a sub-dominant pollen type, bisaccate and inaperturate conifer pollen reaches conspicuous values, and angiosperm pollen exhibits high diversity and significant abundances (i.e., abundances of more than 30% of angiosperm pollen, and a diversity of around 47 taxa in Zone II of the Potomac Group [112]).

Additionally, the late Albian–early Cenomanian palynofloras from the Northern High-latitude Temperate Humid (NHT) belt are characterised by a high diversity of dinocysts, trilete spores and angiosperms (see i.e., [111, 227–229]). For example, the coeval existence of quite different plant communities from those described in eastern Iberia can be regarded if comparing the late Albian floras from Peace River Area (Alberta, Canada) [229]. These floras are characterised by scarce *Classopollis*, high amounts of bisaccate pollen grains of the genera *Alisporites* and *Vitreisporites*, trilete spores such as *Cingutriletes*, *Gleicheniidites*, and *Stereisporites*, as well as conspicuous values of tricolpate angiosperm species (*Cupuliferoidaepollenites parvulus*, *Tricolpites sagax*, *T. vulgaris*).

## Conclusions

The palaeobotanical study of three sections from the Utrillas Group in the Maestrazgo Basin (Iberian Ranges, eastern Spain) reveals the existence of ancient conifer woodlands and fern/angiosperm communities that thrived in the mid-Cretaceous Iberian Desert System. Four vegetation types have been inferred: the first one is characterised by arid woodlands of Cheirolepidiaceae-Cupressaceae, the second is related to coastal settings with Araucariaceae, the third is related to salt marshes, and the fourth represents local fern and angiosperm communities linked to wet interdunes (oases) and wetlands. The existence of ponded areas linked to wet interdunes is interpreted based on the high percentages of fern spores, the common occurrence of angiosperm pollen and the lack of marine palynomorphs. The presence of spikes of *Tucanopollis* spp. and "*Liliacidites*" *minutus* in the CA section might be related to the relevant role of the angiosperms in the colonisation of xeric and disturbed habitats and the ecological successions in arid settings.

The biostratigraphic markers recorded in the studied sections confirm a late Albian age, except for the uppermost levels of the Cortes de Arenoso section (La Hoya amber outcrop), in which an early Cenomanian age might be inferred based on the occurrence of *Echinipollis cenomanensis*, *Polypodiisporonites cenomanianus* and *Senectotetradites grossus*. Although the late Albian palynoflora from the Maestrazgo Basin was influenced by the palaeoclimatic conditions that prevailed in the Northern Hot Arid belt, the studied assemblages can generally be related to others from Europe and North America. However, the presence of pollen grains of the genera *Afropollis*, *Equisetosporites*, *Gnetaceapollenites*, *Steevesipollenites*, *Dichastopollenites* and *Cretacaeiporites* indicates that Iberia was a transitional area between the Northern Gondwanan and the Southern Laurasian provinces. In this sense, the studied macrofloristic assemblages from the Maestrazgo Basin show similarities to others from the European Laurasia.

## Supporting information

**S1 Appendix. Absolute abundances of palynomorphs in the studied sections from the Maestrazgo Basin.**
(XLSX)

**S2 Appendix. List of the identified taxa ordered according their botanical and zoological identification, and the alphabetic citation of the taxonomic authorities.**
(DOC)

## Acknowledgments

The authors would like to thank Dr Rafael López del Valle for the preparation of the amber pieces with arthropod specimens, Dr Juan Pedro Rodríguez-López for his assistance in the fieldwork, Pepa Torres Matilla for her aid with the design of the figures, Dr Xavier Delclòs as supervisor of the PhD Thesis of SÁP, Dr Luis Somoza (CN-IGME CSIC) for the financial support (*AYUDAS EXTRAORDINARIAS MENCIONES EXCELENCIA SEVERO OCHOA del IGME-CSIC*) and the Dirección General de Patrimonio Cultural del Gobierno de Aragón (Spain) for granting the permissions of excavation. We would also like thank the editor (Dr Enrique Peñalver) and the two anonymous referees who provided valuable suggestions for improving the quality of the manuscript.

## Author Contributions

**Conceptualization:** Eduardo Barrón, Daniel Peyrot.

**Data curation:** Eduardo Barrón.

**Formal analysis:** Eduardo Barrón, Daniel Peyrot.

**Funding acquisition:** Eduardo Barrón.

**Investigation:** Eduardo Barrón, Daniel Peyrot, Carlos A. Bueno-Cebollada, Jiří Kvaček, Sergio Álvarez-Parra, Yul Altolaguirre, Nieves Meléndez.

**Methodology:** Eduardo Barrón, Daniel Peyrot, Jiří Kvaček, Sergio Álvarez-Parra, Nieves Meléndez.

**Resources:** Eduardo Barrón.

**Supervision:** Eduardo Barrón.

**Visualization:** Daniel Peyrot, Carlos A. Bueno-Cebollada, Jiří Kvaček, Sergio Álvarez-Parra, Yul Altolaguirre.

**Writing – original draft:** Eduardo Barrón, Daniel Peyrot, Carlos A. Bueno-Cebollada.

**Writing – review & editing:** Eduardo Barrón, Daniel Peyrot, Carlos A. Bueno-Cebollada.

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
