## [Decision Letter · Decision Letter 0]

28 Sep 2022

PONE-D-22-21342Biodiversity of ecosystems in an arid setting: the late Albian plant communities and associated biota from eastern IberiaPLOS ONE

Dear Dr. Barrón,

Thank you for submitting your manuscript to PLOS ONE. After careful consideration, we feel that it has merit but does not fully meet PLOS ONE’s publication criteria as it currently stands. Therefore, we invite you to submit a revised version of the manuscript that addresses the points raised during the review process.

We look forward to receiving your revised manuscript.

Kind regards,

Enrique Peñalver, Ph.D.

Academic Editor

PLOS ONE

Journal Requirements:

 "This study is a contribution to the project CRE CGL2017-84419 AEI/FEDER, UE from the Ministerio de Ciencia, Innovación y Universidades (Spain) and the “Severo Ochoa” extraordinary grants for excellence IGME-CSIC (AECEX2021). The coauthor S.Á.-P. thanks the support from the Secretaria d’Universitats i Recerca de la Generalitat de Catalonia (Spain) and the European Social Fund (2021FI_B2 00003)." 

   "No authors have competing interest"

Additional Editor Comments:

The manuscript needs minor revision. I invite you to submit a revised version of the manuscript that addresses the points raised during the review process. Please, note that Figure 17 is excellent, but the continental aquatic environment that you represent seems very small, but sure that this was very extensive, although most likely in patches, thus I encourage you to extend that aquatic environment to the right of the figure, cutted by the frame, in order to visually express that this figure only shows a small part of the shoreline of a water body much more extensive. 

Reviewers' comments:

Reviewer's Responses to Questions

**Comments to the Author**

1. Is the manuscript technically sound, and do the data support the conclusions?

Reviewer #1: Yes

Reviewer #2: Yes

2. Has the statistical analysis been performed appropriately and rigorously? 

Reviewer #1: Yes

Reviewer #2: No

3. Have the authors made all data underlying the findings in their manuscript fully available?

Reviewer #1: Yes

Reviewer #2: Yes

4. Is the manuscript presented in an intelligible fashion and written in standard English?

Reviewer #1: Yes

Reviewer #2: Yes

5. Review Comments to the Author

Reviewer #1: This is an interesting and well-written MS highlighting and summarizing the Albian-Cenomanian ecosystems of the Maestrazgo Basin, Eastern Spain, integrating palynological and paleobotanical results and placing also the amber deposits and their inclusions in an integrated environmental context. The topic is suitable for the journal and I suggest a minor-moderate revision to make the paper more relevant for a broader readership. The data and figures are all fine but for a journal as PlosOne I would keep the text less detailed and aim at a broader readership. This can be addressed with some minor editing and reformulating of the text.

Abstract: The abstract could be more general pointing out the major conclusions of the paper, especially as there already exist information from the Maestrazgo Basin.

• Point out the integrative work on palynology and paleobotany as this is a new approach

• Highlight your new age determinations of these important amber bearing deposits also in the context of angiosperm radiation. Does the new age assessments have a bearing on the fauna preserved in the amber?

Typo: developped

The general text is well-written and the figures are nice and necessary, however as pointed out above it would benefit from a less specialized style.

Example:

Final sentence of the abstract but this is the case all through the paper:

“Pollen of Afropollis, Dichastopollenites, Cretacaeiporites as well as polyplicate grains produced by ephedroid gymnosperms associate the Iberian ecosystems with the phytogeographic province characterising northern Gondwana.”

This would be a suitable text for a specialist journal such as RPP but for the aimed journal I would suggest to change it to e.g.

Importantly, the studied assemblages include Afropollis, and Cretacaeiporites together with pollen produced by ephedra (known for its tolerance to arid conditions). The presence of these pollen, typical for northern Gondwana, associate the Iberian ecosystems with those characterising northern Gondwana.

Looking forward to see this published!

Reviewer #2: See attachment. See attachment. See attachment. See attachment. See attachment. See attachment. See attachment. See attachment. See attachment. See attachment. See attachment. See attachment. See attachment.

6. PLOS authors have the option to publish the peer review history of their article (what does this mean?). If published, this will include your full peer review and any attached files.

Reviewer #1: No

Reviewer #2: No

---

## [Author Response · Author response to Decision Letter 0]

20 Dec 2022

Dear Dr Enrique Peñalver,

Firstly, we would like to thank you for all the helpful advice and corrections provided with which we fully agree, and we have done our best to meet each and every one of them.

We are also thankful for the constructive comments made by Reviewers 1 and 2 since they have notably improved the quality of our manuscript regarding formal and scientific aspects.

I have listed below the details of how we have tackled and corrected the changes you and both reviewers (1 and 2) kindly suggested according to the different issues you listed in your list of comments. We hope that this new version meets all the requirements you consider necessary for publication.

We look forward to hearing from you and knowing your decision.

Kind regards,

Dr Eduardo Barrón

---

## [Editor Report · Decision Letter 1]

9 Feb 2023

Biodiversity of ecosystems in an arid setting: the late Albian plant communities and associated biota from eastern Iberia

PONE-D-22-21342R1

Dear Dr. Eduardo Barrón,

We’re pleased to inform you that your manuscript has been judged scientifically suitable for publication and will be formally accepted for publication once it meets all outstanding technical requirements.

Kind regards,

Enrique Peñalver, Ph.D.

Academic Editor

PLOS ONE

---

## [Editor Report · Acceptance letter]

16 Feb 2023

PONE-D-22-21342R1 

Biodiversity of ecosystems in an arid setting: the late Albian plant communities and associated biota from eastern Iberia 

Dear Dr. Barrón:

I'm pleased to inform you that your manuscript has been deemed suitable for publication in PLOS ONE. Congratulations! Your manuscript is now with our production department. 

Kind regards, 

on behalf of

Dr. Enrique Peñalver 

Academic Editor

PLOS ONE